# Audio-Visual Dataset Distillation

**Saksham Singh Kushwaha**                    *sakshamsingh.kushwaha@utdallas.edu*
*Department of Computer Science*
*The University of Texas at Dallas*

**Siva Sai Nagender Vasireddy**              *sivasainagender.vasireddy@utdallas.edu*
*Department of Computer Science*
*The University of Texas at Dallas*

**Kai Wang**                                 *kai.wang@comp.nus.edu.sg*
*Institutes of Data Science & School of Computing*
*National University of Singapore*

**Yapeng Tian**                              *yapeng.tian@utdallas.edu*
*Department of Computer Science*
*The University of Texas at Dallas*

**Reviewed on OpenReview:** *https://openreview.net/forum?id=IJlbuSrXmk*

## Abstract

In this article, we introduce *audio-visual dataset distillation*, a task to construct a smaller yet representative synthetic audio-visual dataset that maintains the cross-modal semantic association between audio and visual modalities. Dataset distillation techniques have primarily focused on image classification. However, with the growing capabilities of audio-visual models and the vast datasets required for their training, it is necessary to explore distillation methods beyond the visual modality. Our approach builds upon the foundation of Distribution Matching (DM), extending it to handle the unique challenges of audio-visual data. A key challenge is to jointly learn synthetic data that distills both the modality-wise information and natural alignment from real audio-visual data. We introduce a vanilla audio-visual distribution matching framework that separately trains visual-only and audio-only DM components, enabling us to investigate the effectiveness of audio-visual integration and various multimodal fusion methods. To address the limitations of unimodal distillation, we propose two novel matching losses: implicit cross-matching and cross-modal gap matching. These losses work in conjunction with the vanilla unimodal distribution matching loss to enforce cross-modal alignment and enhance the audio-visual dataset distillation process. Extensive audio-visual classification and retrieval experiments on four audio-visual datasets, AVE, MUSIC-21, VGGSound, and VGGSound-10K, demonstrate the effectiveness of our proposed matching approaches and validate the benefits of audio-visual integration with condensed data. This work establishes a new frontier in audio-visual dataset distillation, paving the way for further advancements in this exciting field. Source code and pre-trained model: `https://github.com/sakshamsingh1/AVDD`.

## 1 Introduction

Dataset distillation aims to learn a condensed dataset such that it retains most of the essential information of the entire training data. Recent progress in dataset distillation techniques, such as gradient matching (Zhao et al., 2020; Zhao & Bilen, 2021), trajectory matching (Cazenavette et al., 2022; Wu et al., 2023; Liu et al., 2023), and distribution matching (Zhao & Bilen, 2023; Zhao et al., 2023; Wang et al., 2022) have

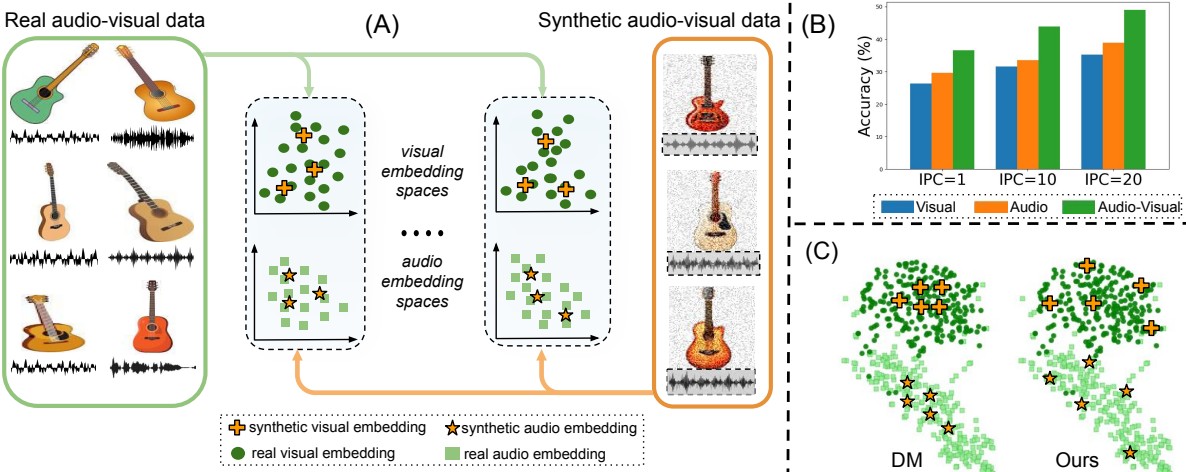

Figure 1: Vanilla audio-visual dataset distillation using distribution matching (A) The synthetic audio and visual data are learned by minimizing the distribution discrepancy between real and synthetic data in these sampled unimodal embedding spaces. (B) Mean test accuracy plot of VGGS-10k dataset for different images per class (IPC) settings. We observe that joint audio-visual can improve performance in data distillation. (C) Feature distribution plot of real audio(■)-visual(●) and synthetic audio(★)-visual(✚) data for vanilla distribution matching(DM) and Our approach. We observe that independently distilled audio-visual synthetic data in DM is pulled towards their modality centers while our additional joint audio-visual losses better cover the real distribution.

achieved remarkable performance on the image datasets. For example, DATM (Guo et al., 2023) achieves lossless distillation using merely 1/5 and 1/10 original sizes of CIFAR100 (Krizhevsky et al., 2009) and TinyImageNet (Le & Yang, 2015) respectively. However, their potential in other domains remains largely underexplored.

With the recent advancements in audio-visual learning (Zhao et al., 2018; Tian et al., 2018; Lin et al., 2019; Gao et al., 2018; Wei et al., 2022), the size of audio-visual datasets (Chen et al., 2020c; Xu et al., 2016; Gemmeke et al., 2017) have significantly increased, which leads to the heavy storage and computational cost of training on these datasets. In this work, we investigate the extension of the dataset distillation to the audio-visual domain. Unlike image distillation (Wang et al., 2018; Cazenavette et al., 2022), audio-visual distillation presents unique challenges: preserving complex cross-modal correlations and addressing the complexities of high-resolution images and the additional audio modality.

Audio-visual integration has proven beneficial for various audio-visual tasks, such as audio-visual event localization (Tian et al., 2018; Wu et al., 2019; Lin et al., 2019), audio-visual sound separation (Zhao et al., 2018; Gao et al., 2018; Tian et al., 2021), and audio-visual action recognition (Kazakos et al., 2019; Zhang et al., 2022; Nagrani et al., 2020). This success raises an important question: *Is audio-visual integration still effective when applied to distilled audio and visual data?* To explore this, we propose *audio-visual dataset distillation*, which aims to learn a smaller, yet representative synthetic audio-visual dataset that is useful for audio-visual learning tasks, distilled from the original large dataset. The task is inherently challenging. An effective audio-visual dataset distillation approach not only needs to condense data from two different modalities, but also preserve the natural cross-modal association between them to ensure the effectiveness of the synthetic multimodal data.

To explore this new problem, we use audio-visual event recognition as the main proxy task and build our approach on top of Distribution Matching (DM) (Zhao & Bilen, 2023). We first propose a vanilla audio-visual distribution matching approach in which we separately train the visual-only and audio-only DM (as shown in Fig. 1A). This vanilla approach allows us to evaluate the effectiveness of audio-visual integration with condensed data and analyze the impact of different multimodal fusion methods on audio-visual event

recognition performance. The results are presented in Fig. 1B, which demonstrates the advantage of joint audio-visual integration over single audio or visual modality in audio-visual dataset distillation setting.

However, distilling audio and visual modalities independently may not effectively capture the natural interplay between audio-visual elements during the condensation from real to synthetic data (as shown in Fig. 1C). And to address this limitation, we further introduce two novel matching losses: implicit cross-matching and cross-modal gap matching. These losses align distributions between synthetic and real data in joint spaces and work in conjunction with vanilla unimodal distribution matching loss in DM to enhance the audio-visual data distillation process. Their combined effect ensures that the synthetic data closely represents the real data, effectively aligning unimodal, cross-modal, and modality gap distributions, resulting in effective audio-visual event recognition and retrieval with the synthetic data. Furthermore, we improve the synthetic data initialization and utilize the high-resolution images (along with audio modality) using herding-based initialization (Welling, 2009) and factor technique (Kim et al., 2022), respectively. The herding aligns the initial synthetic data distribution with the real training data, while the factor method increases the number of features helping overcome the redundancy. Our extensive experiments on four widely used audio-visual datasets: AVE (Tian et al., 2018), MUSIC-21 (Zhao et al., 2019), VGGSound (Chen et al., 2020c), and VGGS-10k (Chen et al., 2020c) support the following findings: effective joint audio-visual integration outperforms unimodal performance in audio-visual dataset distillation, implicit cross-matching and cross-modal gap matching improves the vanilla audio-visual distribution matching by distilling the audio-visual alignment into synthetic data, herding initialization & factor technique further helps improve audio-visual distillation.

The contributions of our work include: 1) a novel audio-visual dataset distillation problem that aims to compress the knowledge of large audio-visual datasets into much smaller synthetic ones. To the best of our knowledge, this is the first work in audio-visual dataset distillation; 2) a systematical investigation into audio-visual dataset distillation to evaluate the efficacy of condensed audio-visual data for audio-visual event recognition; 3) two new audio-visual distribution matching losses that align distributions between synthetic and real data in joint spaces, enforcing cross-modal alignment; and 4) extensive experiments on four audio-visual datasets validating that audio-visual integration with synthetic data is still helpful and our approach can outperform other dataset distillation baselines.

## 2 Related Work

**Audio-Visual Learning.** Videos consist of naturally co-occurring audio and visual signals. To exploit the synchronized and complementary information in the two modalities, several audio-visual learning problems have been explored, such as self-supervised representation learning (Arandjelovic & Zisserman, 2017; 2018; Aytar et al., 2016; Lin et al., 2023; Gong et al., 2022; Owens & Efros, 2018; Hu et al., 2019), audio-visual sound separation (Zhao et al., 2018; Gan et al., 2020; Gao et al., 2018; Tian et al., 2021; Zhou et al., 2020), audio-visual action recognition (Zhang et al., 2022; Lee et al., 2020; Nagrani et al., 2020), audio-visual navigation (Chen et al., 2021; 2020a;b), audio-visual event localization (Tian et al., 2018; Wu et al., 2019; Lin et al., 2019), *etc.* Audio-visual integration has generally demonstrated the ability to improve model performance compared to unimodal models in these tasks. In this work, we investigate the validity of joint audio-visual integration in the context of dataset distillation. We employ audio-visual event recognition (Tian et al., 2018; Lin et al., 2023) and cross-modal retrieval (Surís et al., 2018; Kushwaha & Fuentes, 2023; Wu et al., 2023) as proxy to evaluate the effectiveness of condensed data.

**Dataset Distillation.** The traditional method to reduce the training set size is coreset selection (Castro et al., 2018; Welling, 2009), which heuristically selects a subset of training data. A recent approach, dataset distillation or condensation aims to learn a smaller dataset while still preserving the essential information from the large training dataset. Unlike coreset selection methods, data condensation methods are not limited by the subset of selected real data. It has been shown to benefit several tasks like efficient neural architecture search (Zhao & Bilen, 2023; Zhao et al., 2020), continual learning (Gu et al., 2023; Sangermano et al., 2022; Zhao et al., 2023), federated learning (Xiong et al., 2023; Huang et al., 2023), and privacy preservation (Vinaroz & Park, 2023; Dong et al., 2022). The problem was first introduced in (Wang et al., 2018), which learns condensed data with meta-learning techniques and has since been significantly improved

by more sophisticated techniques like gradient matching (Zhao et al., 2020), trajectory matching (Cazenavette et al., 2022), data parameterization (Kim et al., 2022) and feature alignment (Wang et al., 2022). Most of these methods rely on bi-level optimizations resulting in intensive computation requirements. In contrast, distribution matching (DM) (Zhao & Bilen, 2023) avoids this bi-level optimization and condenses data by matching the feature distribution of real and synthetic data. Recently, Wu et al. (2023) introduced vision-language dataset distillation using joint image-text trajectory matching. However, a straightforward extension of their method to audio-visual distillation is impractical due to differences in modality, task, and scalability. Unlike text, audio is temporally synchronized with visual components, exhibits greater variability, and is typically represented as spectrograms with rich, isolated time-frequency patterns. Additionally, while they focus on instance-based retrieval, our task involves class-based tasks. This difference shifts the focus from learning instance-wise distilled data pairs to class-wise pairing in our approach. Furthermore, their extension of the MTT approach to instance-wise distilled data pairs struggles to scale to higher images-per-class settings in our experiments due to increased memory requirements. In contrast, we extend training-based distillation baselines, moving beyond the coreset selection baselines of Wu et al. (2023). In this work, we advance dataset distillation into audio-visual learning, concentrating on techniques that effectively condense data from both audio and visual modalities while preserving their cross-modal associations.

# 3 Method

## 3.1 Preliminaries

In this section, we describe the problem formulation, introduce the proxy task for evaluating distilled synthetic data, and revisit distribution matching for visual-only dataset distillation.

**Problem Formulation.** Let $x_i^a$ and $x_i^v$ denote the audio waveform and video frame of the $i$-th sample, respectively, with $x_i^{av} = (x_i^a, x_i^v)$ and $y_i$ as the corresponding ground truth category label. Given a large audio-visual training set $\mathcal{T} = \{x_i^{av}, y_i\}_{i=1}^{|\mathcal{T}|}$, our audio-visual dataset distillation task aims to learn a smaller, yet representative synthetic set $\mathcal{S} = \{s_i^{av}, y_i\}_{i=1}^{|\mathcal{S}|}$, where $s_i^{av} = (s_i^a, s_i^v)$. This dataset $\mathcal{S}$, with significantly fewer samples $|\mathcal{S}| \ll |\mathcal{T}|$, should encapsulate the essential information contained in $\mathcal{T}$. The ultimate goal is for models trained on each $\mathcal{T}$ and $\mathcal{S}$ to perform similarly on unseen test data:

$$\mathbb{E}_{(x^{av},y)\sim\mathcal{D}}\left[\ell(\phi_{\theta_{\mathcal{T}}}(x^{av}),y)\right] \simeq \mathbb{E}_{(x^{av},y)\sim\mathcal{D}}\left[\ell(\phi_{\theta_{\mathcal{S}}}(x^{av}),y)\right],$$

where $\mathcal{D}$ is the real test data, $\ell$ is the loss function (i.e. cross-entropy), $\phi_\theta$ is a neural network parameterized by $\theta$, and $\phi_{\theta_{\mathcal{T}}}$ and $\phi_{\theta_{\mathcal{S}}}$ are networks trained on $\mathcal{T}$ and $\mathcal{S}$ respectively.

**Task.** Following image dataset distillation methods (Sachdeva & McAuley, 2023; Yu et al., 2023), we use audio-visual event recognition task as a proxy to investigate the effectiveness of audio-visual dataset distillation. The task involves predicting the event category of a short video clip, characterized by audio waveform $x^a$ and video frame $x^v$. We employ an audio-visual network, illustrated in Fig. 2, to integrate data from both modalities. Specifically, the model uses a visual encoder to extract feature $f^v$ from $x^v$ and an audio encoder extracts feature $f^a$ from the audio spectrogram $m^a$ transformed from input audio $x^a$. These extracted features are then fused to feature $f^{av}$ by a fusion module for predicting the class-event probability $p$. We utilize cross-entropy loss as the objective function $\mathcal{L}_{CE} = -\sum_{i=1}^{|C|} y_i \log(p_i)$, where $|C|$ denotes the total number of event categories. Through this task, we will investigate whether integrating synthetic audio-visual data can enhance recognition in dataset distillation and to what extent different fusion methods affect the model performance.

**Revisit Distribution Matching.** Dataset distillation has demonstrated remarkable success in compressing large image datasets into smaller synthetic ones for visual recognition tasks. Distribution matching (DM) (Zhao & Bilen, 2023) stands out as one of the prominent approaches, which aims to generate synthetic data that closely resembles the distribution of real samples in the feature space. This is achieved by minimizing the feature distance between the distributions of real and

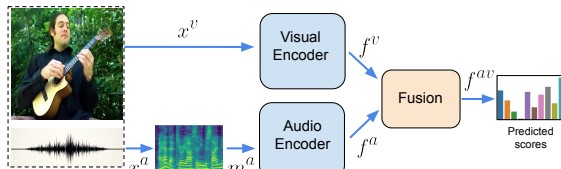

Figure 2: Audio-visual event recognition.

feature space. This is achieved by minimizing the feature distance between the distributions of real and

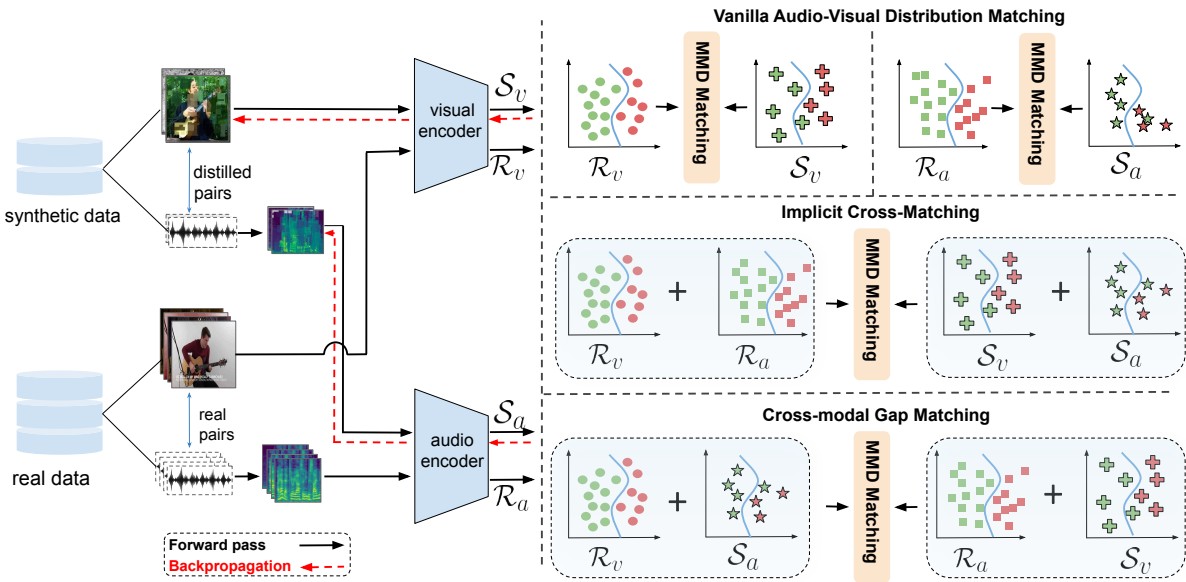

Figure 3: The proposed audio-visual dataset distillation framework. The synthetic audio-visual feature distribution and real audio-visual feature distribution are matched using three main components: Vanilla audio-visual distribution matching, Implicit Cross-Matching (ICM), and Cross-modal Gap Matching (CGM). The vanilla distribution matching loss ensures the alignment of the same modality matching while ICM and CGM facilitate cross-modal matching. $\mathcal{R}_a$, $\mathcal{R}_v$, $\mathcal{S}_a$, $\mathcal{S}_v$ are the real audio, real visual, synthetic audio, synthetic visual distribution respectively. Maximum Mean discrepancy (MMD) (Gretton et al., 2012) is used as a measurement for distribution matching. The distilled image and audio spectrogram pairs are shown at an intermediate state and hence slightly modified from the initialized real data.

synthetic samples, ensuring that the synthetic data effectively captures the essential characteristics of the original dataset. Specifically, it uses randomly initialized neural networks as feature extractors and minimizes the spatial distribution using an empirical estimate of maximum mean discrepancy (MMD) (Gretton et al., 2012). DM maps each training image $x^v \in \mathbb{R}^d$ to a lower dimensional space using a family of parametric functions $\psi_{\theta_v} : \mathbb{R}^d \to \mathbb{R}^{d'}$ where $d' \ll d$. Here, $\psi_{\theta_v}$ can be implemented using neural networks with random weights. It also augments data by applying differential Siamese augmentation $\mathcal{A}_\omega(\cdot)$ (Zhao & Bilen, 2021) to real and synthetic data, where $\omega \sim \Omega$ is the augmentation parameter. To this end, we will solve the following optimization problem:

$$\min_{S_v} \mathbb{E}_{\substack{\omega \sim \Omega \\ \theta_v \sim P_{\theta_v}}} \left\| \frac{1}{|\mathcal{T}_v|} \sum_{i=1}^{|\mathcal{T}_v|} \psi_{\theta_v}(\mathcal{A}_\omega(\mathbf{x}_i^v)) - \frac{1}{|S_v|} \sum_{j=1}^{|S_v|} \psi_{\theta_v}(\mathcal{A}_\omega(\mathbf{s}_j^v)) \right\|^2 \quad (1)$$

where $P_{\theta_v}$ is the distribution of network parameters. By minimizing the discrepancy between two distributions in various embedding spaces by sampling $\theta_v$, we can learn the synthetic visual data $S_v$. Upon generating the synthetic dataset $S_v$, we will use it as training data to train our visual recognition model, optimizing it with the cross-entropy loss. This trained model will then predict class labels for real test image samples.

## 3.2 Audio-Visual Dataset Distillation

In this section, we will explore dataset distillation approaches capable of simultaneously learning synthetic audio and visual data for audio-visual event recognition. We will first introduce a vanilla model that extends visual distribution matching and then present two novel approaches tailored to this task, utilizing joint distribution matching. Our approach is illustrated in Fig. 3.

**Vanilla Audio-Visual Distribution Matching.** We build our audio-visual dataset distillation approach on the top of DM. To tackle the new multimodal task, one naive approach is to use DM to condense audio

and visual data separately during data distillation and then combine the trained distilled audio and visual data for audio-visual event recognition. Alongside a visual-only DM loss $\mathcal{L}_{base}^v$ as detailed in Eq. 1, we introduce a DM loss[1] for the audio modality:

$$\mathcal{L}_{base}^a = ||\frac{1}{|\mathcal{T}_a|}\sum_{i=1}^{|\mathcal{T}_a|}\psi_{\theta_a}(\mathcal{A}_\omega(\mathbf{x}_i^a)) - \frac{1}{|\mathcal{S}_a|}\sum_{j=1}^{|\mathcal{S}_a|}\psi_{\theta_a}(\mathcal{A}_\omega(\mathbf{s}_j^a))||^2, \tag{2}$$

where $\psi_{\theta_a}$ denotes randomly initialized audio network. The vanilla approach to audio-visual distribution matching optimizes the following objective function:

$$\mathcal{L}_{base}^{av} = \mathcal{L}_{base}^a + \mathcal{L}_{base}^v. \tag{3}$$

However, distilling modalities separately fails to capture the natural interplay between real audio-visual data, so we introduce two joint matching losses to better align the distilled synthetic data.

**Distribution Matching Between Joint Spaces.** To enforce cross-modal alignment during distribution matching, one straight-forward approach is to introduce a cross-modal distribution matching loss. This loss function will minimize the discrepancy between the distributions of real audio data and synthetic visual data, as well as between real visual data and synthetic audio data. However, this approach has limitations due to the use of randomly initialized models $\psi_{\theta_v}$ and $\psi_{\theta_a}$ for mapping audio and visual data into separate embedding spaces in DM. As shown in Fig. 4, there exists a strong modality gap between the two modalities. Since only the synthetic data is learnable, real data remains fixed, and model weights are randomly initialized and not trainable, directly matching one modality to another cannot mitigate the modality gap and can lead to instability during training (refer to Appendix A for a detailed explanation of the failure exploration). This is in contrast to real data, where audio-visual associations naturally exist. To address these challenges, we propose two new learning losses that encourage cross-modal alignment in a joint embedding space, rather than merely performing cross-modal matching in individual unimodal audio and visual spaces.

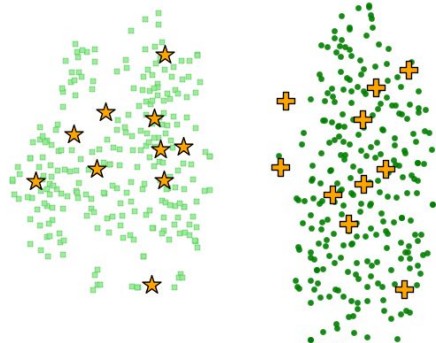

Figure 4: Feature distribution plot of real audio(■)-visual(●) and synthetic audio(★)-visual(✚) data at an intermediate training stage of the vanilla audio-visual distribution matching. The audio and visual features are extracted from randomly initialized audio and visual networks, respectively. We can observe the strong modality gap between the two modalities.

**Implicit Cross-matching (ICM).** Given the inherent challenges of achieving separate cross-modal distribution matching, we propose a novel approach that bypasses the difficulty by introducing a joint audio-visual distribution matching loss. This loss function effectively aligns the joint audio-visual distributions between real ($\mathcal{D}^r$) and synthetic ($\mathcal{D}^s$) data, enabling implicit cross-modal distribution matching. The loss is formally defined for each class as follows:

$$\mathcal{D}^r = \bar{R}^a + \bar{R}^v = \left[\frac{1}{|\mathcal{T}_a|}\sum_{i=1}^{|\mathcal{T}_a|}\psi_{\theta_a}(\mathcal{A}_\omega(\mathbf{x}_i^a)) + \frac{1}{|\mathcal{T}_v|}\sum_{i=1}^{|\mathcal{T}_v|}\psi_{\theta_v}(\mathcal{A}_\omega(\mathbf{x}_i^v))\right], \tag{4}$$

$$\mathcal{D}^s = \bar{S}^a + \bar{S}^v = \left[\frac{1}{|\mathcal{S}_a|}\sum_{j=1}^{|\mathcal{S}_a|}\psi_{\theta_a}(\mathcal{A}_\omega(\mathbf{s}_j^a)) + \frac{1}{|\mathcal{S}_v|}\sum_{j=1}^{|\mathcal{S}_v|}\psi_{\theta_v}(\mathcal{A}_\omega(\mathbf{s}_j^v))\right], \tag{5}$$

$$\mathcal{L}_{ICM}^{av} = ||\mathcal{D}^r - \mathcal{D}^s||^2, \tag{6}$$

Optimizing this loss term effectively compels our model to learn synthetic audio-visual data $\{\mathcal{S}_a, \mathcal{S}_v\}$ that closely resembles and represents the real dataset $\{\mathcal{T}_a, \mathcal{T}_v\}$. Here, the loss term in Eq. 6 can be re-written as $\mathcal{L}_{ICM}^{av} = ||(\bar{R}^a + \bar{R}^v) - (\bar{S}^a + \bar{S}^v)||^2 = ||(\bar{R}^a - \bar{S}^v) + (\bar{R}^v - \bar{S}^a)||^2 \leq ||(\bar{R}^a - \bar{S}^v)||^2 + ||(\bar{R}^v - \bar{S}^a)||^2$. This formulation reveals that the loss implicitly enforces cross-modal matching between real audio and synthetic visual data, as well as between real visual and synthetic audio data.

---

[1]For simplicity, we omit the $\mathbb{E}$ term in the loss.

**Cross-modal Gap Matching (CGM).** Besides the ICM loss, we introduce further constraints to align the distributions of ($\bar{R}_a$ and $\bar{S}_v$) and ($\bar{R}_v$ and $\bar{S}_a$), as follows:

$$\mathcal{D}^{av} = \left[ \frac{1}{|\mathcal{T}_a|} \sum_{i=1}^{|\mathcal{T}_a|} \psi_{\theta_a}(\mathcal{A}_\omega(\mathbf{x}_i^a)) + \frac{1}{|\mathcal{S}_v|} \sum_{j=1}^{|\mathcal{S}_v|} \psi_{\theta_v}(\mathcal{A}_\omega(\mathbf{s}_j^v)) \right] \tag{7}$$

$$\mathcal{D}^{va} = \left[ \frac{1}{|\mathcal{T}_v|} \sum_{i=1}^{|\mathcal{T}_v|} \psi_{\theta_v}(\mathcal{A}_\omega(\mathbf{x}_i^v)) + \frac{1}{|\mathcal{S}_a|} \sum_{j=1}^{|\mathcal{S}_a|} \psi_{\theta_a}(\mathcal{A}_\omega(\mathbf{s}_j^a)) \right] \tag{8}$$

$$\mathcal{L}_{MGM}^{av} = ||\mathcal{D}^{av} - \mathcal{D}^{va}||^2 \tag{9}$$

This addition ensures that the synthetic data closely represent the corresponding real data without misaligning the existing matches of $\bar{S}^a \leftrightarrow \bar{R}^a$ and $\bar{S}^v \leftrightarrow \bar{R}^v$ enforced in unimodal DM and Joint Matching. With a simple re-writing, we can obtain $\mathcal{L}_{ICM}^{av} = ||(\bar{R}^a + \bar{S}^v) - (\bar{R}^v + \bar{S}^a)||^2 = ||(\bar{R}^a - \bar{R}^v) - (\bar{S}^a - \bar{S}^v)||^2$. We can see that it will help to align the modality gap between real and synthetic data to strengthen the joint audio-visual distribution matching.

**Final Loss.** For training, we will jointly optimize the three loss terms:

$$\mathcal{L}_{final}^{av} = \mathcal{L}_{base}^{av} + \lambda_{ICM} \cdot \mathcal{L}_{ICM}^{av} + \lambda_{CGM} \cdot \mathcal{L}_{CGM}^{av}. \tag{10}$$

Here $\lambda_{ICM}$ and $\lambda_{CGM}$ are the weights for implicit cross-matching and cross-modal gap matching losses respectively. These three loss terms work collaboratively to enhance the audio-visual dataset distillation process. Their combined effect ensures that the synthetic data closely corresponds to the real data, aligning unimodal, cross-modal, and modality gap distributions effectively.

**Training algorithm.** We train the synthetic data for K iterations and for each iteration, we sample random model $\psi_{\theta_a}$ and $\psi_{\theta_v}$ for audio and visual embedding. We then randomly sample real audio-visual data batch and audio-visual synthetic data batch and augmentation parameter for each class. We calculate the mean discrepancy between each modality individually and implicit cross-matching and cross-modal gap-matching representations for each class and then sum them as loss $\mathcal{L}$. We update the synthetic data by backpropagating $\mathcal{L}$ for each class with learning rate $\eta$. The overall training algorithm is shown in Appendix Algorithm 1.

**Improved initialization and storage.** Herding (Welling, 2009) is a coreset selection method that greedily selects data points to minimize the coreset center and the original data center. It has shown superior performance among the coreset selection methods in several previous dataset distillation research (Sachdeva & McAuley, 2023; Yu et al., 2023). We adopt herding as it aligns the initial synthetic data distribution with the real data, offering an advantage over random initialization. The factor technique (Zhang et al., 2023; Kim et al., 2022; Zhao et al., 2023) aims to increase the number of representative features extracted from synthetic data without any additional cost. Specifically, given a factor parameter $l$ each synthetic image is factored into $l^2$ mini-examples and then up-sampled to its original size in training. By combining herding initialization and factor technique, we can further improve our joint audio-visual distribution matching. (More details are provided in Appendix B)

# 4 Experiments

Following the evaluation protocols from previous dataset distillation studies (Zhao & Bilen, 2023; Cazenavette et al., 2022), we use audio-visual event recognition as the main proxy task to assess the classification accuracy on held-out test data of deep networks trained from scratch on our distilled audio and visual data.

## 4.1 Experimental Settings

**Datasets.** We use four widely used audio-visual datasets to validate distillation methods. Each sample represents a one-second video clip, comprising a center frame and its corresponding audio.

*VGGSound & VGGS-10k*: VGGSound (Chen et al., 2020c) is a large scale audio-visual dataset consisting of around 200k YouTube videos from 309 classes. We select the center one-second video from each original

Table 1: Recognition results with synthetic audio (A), visual (V), and audio-visual (AV) data on VGGS-10K. We observe that audio-visual modeling generally outperforms individual ones, highlighting the necessity of audio-visual data distillation.

| | Coreset Selection | | | | | | Training Set Synthesis | | | | | | Whole data | | |
| | Random | | | Herding[1] | | | MTT[2] | | | DM[3] | | | | | |
| IPC | A | V | AV | A | V | AV | A | V | AV | A | V | AV | A | V | AV |
|---|---|---|---|---|---|---|---|---|---|---|---|---|---|---|---|
| 1 | $14.27_{\pm0.97}$ | $11.65_{\pm1.45}$ | $\mathbf{15.44}_{\pm1.87}$ | $\mathbf{26.32}_{\pm1.57}$ | $14.72_{\pm2.87}$ | $20.77_{\pm2.77}$ | $30.99_{\pm1.48}$ | $24.15_{\pm2.25}$ | $\mathbf{34.13}_{\pm3.62}$ | $29.60_{\pm2.33}$ | $26.40_{\pm1.10}$ | $\mathbf{36.54}_{\pm2.52}$ | | | |
| 10 | $32.01_{\pm1.64}$ | $22.71_{\pm1.57}$ | $\mathbf{32.50}_{\pm2.03}$ | $34.58_{\pm1.98}$ | $28.9_{\pm1.44}$ | $\mathbf{39.89}_{\pm1.64}$ | $36.57_{\pm2.57}$ | $25.41_{\pm1.58}$ | $\mathbf{36.79}_{\pm1.97}$ | $33.60_{\pm1.35}$ | $31.63_{\pm1.96}$ | $\mathbf{43.85}_{\pm1.75}$ | $62.07_{\pm0.54}$ | $48.19_{\pm0.54}$ | $\mathbf{68.24}_{\pm0.75}$ |
| 20 | $36.78_{\pm2.88}$ | $31.05_{\pm1.17}$ | $\mathbf{45.10}_{\pm2.31}$ | $44.11_{\pm1.47}$ | $34.58_{\pm0.84}$ | $\mathbf{50.20}_{\pm0.74}$ | $45.73_{\pm1.03}$ | $29.52_{\pm1.43}$ | $\mathbf{51.87}_{\pm1.26}$ | $38.93_{\pm3.52}$ | $35.23_{\pm1.16}$ | $\mathbf{49.01}_{\pm2.44}$ | | | |

[1] (Welling, 2009), [2] (Cazenavette et al., 2022), [3] (Zhao & Bilen, 2023)

Table 2: Audio-visual event recognition results for different fusion methods and images per class (IPC) on VGGS-10K. The results are presented over synthetic audio and visual data distilled using audio-only and visual-only DM (Zhao & Bilen, 2023), respectively. For fixed IPC, the same synthetic data is used. Ensemble consistently achieves the highest accuracy.

| | | Only-A | Only-V | Audio-Visual Fusion | | | |
| | | | | Concat | Sum | Attention | Ensemble |
|---|---|---|---|---|---|---|---|
| | 1 | $29.60_{\pm2.33}$ | $26.40_{\pm1.10}$ | $33.77_{\pm1.65}$ | $34.72_{\pm1.27}$ | $9.97_{\pm0.83}$ | $\mathbf{36.54}_{\pm2.52}$ |
| IPC | 10 | $33.60_{\pm1.35}$ | $31.63_{\pm1.96}$ | $41.71_{\pm1.27}$ | $40.49_{\pm1.83}$ | $10.11_{\pm0.35}$ | $\mathbf{43.85}_{\pm1.75}$ |
| | 20 | $38.93_{\pm3.52}$ | $35.23_{\pm1.16}$ | $46.59_{\pm1.34}$ | $46.05_{\pm1.74}$ | $11.10_{\pm1.88}$ | $\mathbf{49.01}_{\pm2.44}$ |

clip of the train/test split and have around 165k/13k samples respectively. For exploratory analyses and experimental setup of this novel task, we randomly selected a subset of 10 classes from VGGSound with 8808 train videos and 444 test videos. This subset is referred to as VGGS-10k.

*MUSIC-21*: (Zhao et al., 2019) comprises synchronized audio-visual recordings featuring 21 distinct musical instruments. For our study, we focus exclusively on the solo performances subset and segment each video clip into discrete, non-overlapping windows of one second. We randomly partition this subset into train/val/test splits of 146,908/7,103/42,440 samples, respectively.

*AVE*: (Tian et al., 2018) consists of 4,143 video clips spanning over 28 event categories. We segment each clip into non-overlapping one-second windows aligned with the synchronized annotations, resulting in train/val/test splits of 27,726/3,288/3,305 samples, respectively.

**Implementation Details.** Following previous distillation methods (Zhao et al., 2023; Zhao & Bilen, 2023; Cazenavette et al., 2022), we use a ConvNet architecture (Zhao et al., 2020) for both audio and visual inputs. The audio ConvNet consists of 3 blocks with convolution, normalization, ReLU, and pooling layers. For larger image inputs ($224 \times 224$), we use 5 such blocks. We use a learning rate of 0.2 and an SGD optimizer with a momentum of 0.5. Our synthetic data is initialized with Herding-selected audio-visual (AV) pairs and trained with a batch size of 128. For IPC 1 and 10, we set $\lambda_{ICM}$ and $\lambda_{CGM}$ to 10, and for IPC 20, we set them to 20. Audio is sampled at 11kHz and transformed into $128 \times 56$ log mel-spectrograms.

*Baselines.* For DM (Zhao & Bilen, 2023), we use the same learning rate, optimizer, and AV pair batch size. For DC (Zhao et al., 2020), DSA (Zhao & Bilen, 2021), and MTT (Cazenavette et al., 2022), we extend image-only distillation to independently learn audio and visual distilled data. We randomly select AV pairs for the random selection method. For Herding, we greedily select samples closest to the cluster center and follow (Wu et al., 2023) to get AV data by concatenating AV features extracted from models trained on the whole dataset. More details are in Appendix C.

**Evaluation.** We report the mean accuracy and standard deviation of 3 runs where the model is randomly initialized and trained for 30 epochs using the learned synthetic data. Each run consists of 5000 iterations and we use a similar setup as (Tian & Xu, 2021) to train the audio-visual event recognition models. For reference, we also report the performance of models trained on the whole dataset under the same training conditions.

Table 3: Comparison with previous data distillation methods for audio-visual event recognition. Following common practice, we evaluate our method on four different datasets with different numbers of synthetic images per class(IPC). Ratio(%): the ratio of condensed images to the whole training set. Whole Data: the accuracy of the ConvNet model trained on the whole training set and is the upper bound of the performance. '-' refers to configurations for which the method couldn't scale up.

| | IPC | Ratio% | Coreset Selection | | Training Set Synthesis | | | | | Whole data |
| | | | Random | Herding[1] | DC[2] | DSA[3] | MTT[4] | DM[5] | **Ours** | |
|---|---|---|---|---|---|---|---|---|---|---|
| VGGS-10K | 1 | 0.11 | $15.44_{\pm1.87}$ | $20.77_{\pm2.11}$ | $18.28_{\pm1.36}$ | $19.32_{\pm1.35}$ | $34.13_{\pm3.62}$ | $36.54_{\pm2.52}$ | $\mathbf{40.41}_{\pm1.81}$ | |
| | 10 | 1.13 | $32.01_{\pm1.64}$ | $39.89_{\pm1.64}$ | $32.10_{\pm0.84}$ | $36.61_{\pm1.04}$ | $36.79_{\pm1.97}$ | $43.85_{\pm1.75}$ | $\mathbf{54.99}_{\pm1.73}$ | $68.24_{\pm0.75}$ |
| | 20 | 2.26 | $45.1_{\pm2.31}$ | $50.2_{\pm0.74}$ | - | - | $51.87_{\pm1.26}$ | $49.01_{\pm2.44}$ | $\mathbf{58.04}_{\pm1.68}$ | |
| VGGSound | 1 | 0.18 | $1.38_{\pm0.12}$ | $2.14_{\pm0.17}$ | - | - | $1.55_{\pm0.15}$ | $3.08_{\pm0.21}$ | $\mathbf{4.97}_{\pm0.30}$ | |
| | 10 | 1.87 | $5.55_{\pm0.19}$ | $7.09_{\pm0.09}$ | - | - | - | $6.40_{\pm0.28}$ | $\mathbf{8.23}_{\pm0.24}$ | $25.54_{\pm0.19}$ |
| | 20 | 3.74 | $8.00_{\pm0.14}$ | $9.64_{\pm0.14}$ | - | - | - | $8.64_{\pm0.14}$ | $\mathbf{9.85}_{\pm0.34}$ | |
| MUSIC-21 | 1 | 0.014 | $24.12_{\pm4.33}$ | $26.15_{\pm2.01}$ | $22.60_{\pm1.13}$ | $22.98_{\pm1.16}$ | $28.71_{\pm1.23}$ | $38.26_{\pm1.32}$ | $\mathbf{44.02}_{\pm2.21}$ | |
| | 10 | 0.14 | $45.77_{\pm1.74}$ | $51.89_{\pm1.39}$ | - | - | $42.25_{\pm1.07}$ | $54.78_{\pm1.39}$ | $\mathbf{68.07}_{\pm0.98}$ | $85.93_{\pm0.084}$ |
| | 20 | 0.28 | $54.86_{\pm1.85}$ | $59.98_{\pm0.85}$ | - | - | - | $61.06_{\pm1.31}$ | $\mathbf{70.30}_{\pm0.69}$ | |
| AVE | 1 | 0.10 | $10.07_{\pm1.16}$ | $11.84_{\pm0.4}$ | $10.45_{\pm0.39}$ | $10.76_{\pm0.62}$ | $12.13_{\pm0.41}$ | $21.70_{\pm1.46}$ | $\mathbf{23.00}_{\pm1.37}$ | |
| | 10 | 1.0 | $20.0_{\pm1.45}$ | $26.86_{\pm0.52}$ | - | - | $23.15_{\pm0.95}$ | $28.14_{\pm1.80}$ | $\mathbf{36.82}_{\pm0.88}$ | $52.20_{\pm0.38}$ |
| | 20 | 2.0 | $26.32_{\pm1.01}$ | $33.04_{\pm0.38}$ | - | - | - | $32.57_{\pm0.97}$ | $\mathbf{40.13}_{\pm1.00}$ | |

[1] (Welling, 2009), [2] (Zhao et al., 2020), [3] (Zhao & Bilen, 2021), [4] (Cazenavette et al., 2022), [5] (Zhao & Bilen, 2023)

## 4.2 Experimental Results

**Audio-visual integration with distilled data is still helpful.** Audio-visual integration has consistently demonstrated superior performance over unimodal data across various tasks for real data. We aim to further investigate whether this advantage extends to audio-visual distilled data for different data distillation approaches. To explore this, we employed audio-visual event recognition as a benchmark to compare the performance of unimodal distilled data and audio-visual distilled data. Using the VGGS-10K dataset and an ensemble model trained on individually learned audio-visual data, we evaluate several data distillation methods and different synthetic data sizes. The results, shown in Tab. 1, clearly demonstrate that audio-visual integration consistently outperforms unimodal modalities in most cases. This observation suggests that effective audio-visual integration remains beneficial even for distilled data.

**Multimodal Fusion.** Audio-Visual fusion plays a crucial role in the performance of multimodal models. We investigated whether different audio-visual fusion strategies influence the performance of models trained on distilled synthetic data. To address this question, we compared several standard audio-visual fusion approaches: Sum, Concatenation (Concat), Ensemble, and audio-guided visual attention (Attention) (Pian et al., 2023). Tab. 2 presents the audio-visual recognition accuracy across different synthetic data size settings using separately learned VGGS-10K data with DM. From Tab. 2, we can see that the ensemble method consistently outperforms other approaches in all image-per-class settings, followed by a comparable performance of Concatenation and Sum, and the lowest performance of attention fusion. The comparatively low fusion results in attention fusion can be accounted for classwise alignment losses, spatial distortions (as visualized in Fig. 5), and a larger number of trainable model parameters (More details are in the Appendix). Consequently, we employ the ensemble fusion as the default. Furthermore, these results can also demonstrate that audio-visual integration with synthetic data is still helpful when employing different fusion methods.

**Comparison with Data Distillation Baselines.** Tab. 3 compares the performance of our audio-visual distillation method with other baselines across four datasets and three images-per-class (IPC) values. Our audio-visual data distillation approach consistently outperforms vanilla audio-visual distillation with DM, demonstrating the effectiveness of incorporating joint matching losses to strengthen cross-modal alignment. In addition, similar to previous image-only distillation methods (Cazenavette et al., 2022), we observe diminishing returns as IPC increases. For instance, in MUSIC-21, there's a significant performance jump from 44.02% to 68.07% when moving from 1 to 10 IPC, while the improvement from 10 to 20 IPC is more modest, reaching 70.30%. Interestingly, a simple heuristic method like herding becomes competitive starting at IPC 10 and may even outperform DM and MTT in some cases. This could be attributed to the increase in

Table 4: Ablation study at IPC=10 for (left) proposed losses and (right) herding and factor.

| Random | Herding | Factor | Base | ICM | CGM | VGGS-10k | AVE |
|---|---|---|---|---|---|---|---|
| ✓ | | | | | | $32.01_{\pm1.64}$ | $20.00_{\pm1.45}$ |
| | ✓ | | | | | $39.89_{\pm1.64}$ | $26.86_{\pm0.52}$ |
| | ✓ | ✓ | | | | $40.28_{\pm2.34}$ | $31.80_{\pm1.28}$ |
| | ✓ | ✓ | ✓ | | | $45.31_{\pm2.68}$ | $34.80_{\pm1.68}$ |
| | ✓ | ✓ | | ✓ | | $45.73_{\pm2.99}$ | $35.04_{\pm0.90}$ |
| | ✓ | ✓ | | | ✓ | $45.18_{\pm2.80}$ | $34.67_{\pm0.95}$ |
| | ✓ | ✓ | ✓ | ✓ | | $49.07_{\pm1.97}$ | $35.13_{\pm1.14}$ |
| | ✓ | ✓ | ✓ | | ✓ | $49.16_{\pm1.22}$ | $35.51_{\pm0.78}$ |
| | ✓ | ✓ | | ✓ | ✓ | $49.48_{\pm1.16}$ | $35.43_{\pm1.20}$ |
| | ✓ | ✓ | ✓ | ✓ | ✓ | $\mathbf{54.99}_{\pm1.73}$ | $\mathbf{36.82}_{\pm0.88}$ |

| Random | Herding | Factor | Base | ICM + CGM | VGGS-10k | AVE |
|---|---|---|---|---|---|---|
| ✓ | | | ✓ | | $43.85_{\pm1.75}$ | $28.14_{\pm1.80}$ |
| ✓ | | | ✓ | ✓ | $45.87_{\pm2.11}$ | $30.71_{\pm1.17}$ |
| | ✓ | | ✓ | | $44.47_{\pm1.96}$ | $32.36_{\pm0.72}$ |
| | ✓ | | ✓ | ✓ | $46.68_{\pm2.27}$ | $33.60_{\pm0.66}$ |
| ✓ | | ✓ | ✓ | | $41.67_{\pm1.03}$ | $31.51_{\pm0.50}$ |
| ✓ | | ✓ | ✓ | ✓ | $53.41_{\pm1.56}$ | $35.12_{\pm0.80}$ |
| | ✓ | ✓ | ✓ | | $45.31_{\pm2.68}$ | $34.80_{\pm1.68}$ |
| | ✓ | ✓ | ✓ | ✓ | $54.99_{\pm1.73}$ | $36.82_{\pm0.88}$ |

Figure 6: t-SNE distribution plot of synthetic audio-visual data (IPC=10) learned by DM and Ours($l$=1), with same initilization. (green, blue), (red, **black**) and (purple, yellow) points are the real (audio, visual) points for the first 3 classes of VGGS-10k. The synthetic (audio, visual) data is represented by (★, ▲). We observe that our synthetic audio and visual distributions better resemble the real distributions.

the number of learning parameters due to the high-resolution images. Scaling in DM proved more feasible compared to MTT, DC, and DSA. In fact, fitting visual-only distillation using MTT was not possible even with minimal configurations. This is due to large memory consumption when unrolling optimization through SGD steps Cui et al. (2022); Zhou et al. (2022); Cui et al. (2023) for large datasets and higher IPC.

**Ablation Study.** We developed two novel audio-visual distribution matching losses aimed at more effectively distilling essential audio-visual correlations into synthetic data. Additionally, we also use herding initialization and factor technique. To validate the contributions of these components, we conducted an ablation study by systematic addition. The ablation study results on VGGS-10K and AVE are shown in Tab. 4, from which we can see that each of the proposed parts has a positive influence on the final results. These findings unequivocally demonstrate the effectiveness of our proposed ICM and CGM losses and herding initialization and factor technique in helping the audio-visual data distillation process.

**Visualization.** To showcase our distilled data, we plot the learned audio and visual data in Fig. 5 at different IPCs. We observe that with an increased IPC, the synthetic data remains perceptually closer to the original audio-visual sample. The artifacts/patches in the synthetic data arise as the approach distills information from the entire training data into a few synthetic samples. In addition, the spectrogram and image pixels are learnable parameters, making artifacts more prominent in settings with smaller IPCs. Despite this, the model achieves high accuracy on unseen real test data, indicating that these artifacts do not negatively impact performance. Furthermore, we compare the data distribution of the first three classes of VGGS-10k in Fig. 6. It illustrates how our approach better captures the underlying distribution of the real data.

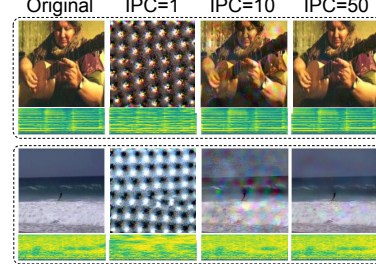

Figure 5: Visualization of distilled data under different IPCs.

**Cross-architecture performance.** In Tab. 5, we evaluate how well our distilled data performs on unseen architectures. We observe that our method outperforms DM and MTT on all the unseen architectures by a large margin, thus highlighting its superior generalization ability.

Table 5: Cross-architecture performance for distilled data on VGGS-10k and IPC=1.

| | Evaluation Architectures | | | | |
|---|---|---|---|---|---|
| | ConvNet | VGG11 | LeNet | ResNet18 | AlexNet |
| MTT | $34.13_{\pm3.62}$ | $30.16_{\pm1.30}$ | $23.93_{\pm3.14}$ | $24.65_{\pm1.53}$ | $22.62_{\pm1.39}$ |
| DM | $36.54_{\pm2.52}$ | $26.95_{\pm3.59}$ | $26.14_{\pm2.66}$ | $21.99_{\pm1.03}$ | $26.68_{\pm1.82}$ |
| Ours | $\mathbf{40.41}_{\pm1.81}$ | $\mathbf{34.00}_{\pm1.81}$ | $\mathbf{32.37}_{\pm5.23}$ | $\mathbf{30.48}_{\pm1.98}$ | $\mathbf{31.69}_{\pm2.72}$ |

Table 6: Audio-visual retrieval results on four different datasets. We train audio-visual ConvNet networks with shared classifier and using Random, DM and Our distilled data of IPC 20. Whole data is trained using the entire training data. We observe that our approach helps to distill better audio-visual alignment.

| | Method | VGGS-10k test subset | | | VGGSound test subset | | | Music-21 test subset | | | AVE test subset | | |
|---|---|---|---|---|---|---|---|---|---|---|---|---|---|
| | | R@1↑ | R@5↑ | MedR↓ | R@1↑ | R@5↑ | MedR↓ | R@1↑ | R@5↑ | MedR↓ | R@1↑ | R@5↑ | MedR↓ |
| A→V | Random | $13.33_{\pm5.03}$ | $52.00_{\pm14.00}$ | $5.83_{\pm1.75}$ | $0.69_{\pm0.32}$ | $3.26_{\pm0.88}$ | $123.0_{\pm1.73}$ | $15.55_{\pm1.09}$ | $34.92_{\pm2.39}$ | $9.33_{\pm0.57}$ | $7.62_{\pm3.21}$ | $30.23_{\pm4.06}$ | $12.33_{\pm2.30}$ |
| | DM(Zhao & Bilen, 2023) | $8.66_{\pm1.15}$ | $47.33_{\pm5.77}$ | $6.66_{\pm1.52}$ | $0.82_{\pm0.35}$ | $3.88_{\pm0.26}$ | $113.00_{\pm6.24}$ | $16.82_{\pm2.39}$ | $37.14_{\pm1.90}$ | $9.00_{\pm1.00}$ | $6.90_{\pm2.29}$ | $32.14_{\pm0.71}$ | $11.16_{\pm1.75}$ |
| | **Ours** | $\mathbf{19.33}_{\pm2.30}$ | $\mathbf{59.33}_{\pm1.15}$ | $\mathbf{3.66}_{\pm0.57}$ | $\mathbf{0.90}_{\pm0.28}$ | $\mathbf{4.1}_{\pm0.35}$ | $\mathbf{110}_{\pm5.19}$ | $\mathbf{31.74}_{\pm0.54}$ | $\mathbf{53.01}_{\pm1.98}$ | $\mathbf{5.00}_{\pm0.00}$ | $\mathbf{13.09}_{\pm2.88}$ | $\mathbf{35.00}_{\pm1.88}$ | $\mathbf{9.00}_{\pm0.00}$ |
| | Whole data | $44.00_{\pm2.00}$ | $74.00_{\pm5.03}$ | $2.00_{\pm0.00}$ | $1.79_{\pm0.73}$ | $7.81_{\pm1.30}$ | $70.33_{\pm3.51}$ | $57.14_{\pm2.51}$ | $78.41_{\pm2.85}$ | $1.00_{\pm0.00}$ | $27.61_{\pm5.35}$ | $51.66_{\pm4.06}$ | $4.66_{\pm1.15}$ |
| V→A | Random | $10.66_{\pm2.30}$ | $49.33_{\pm5.77}$ | $6.00_{\pm0.86}$ | $0.79_{\pm0.27}$ | $3.60_{\pm0.54}$ | $160.66_{\pm2.64}$ | $10.16_{\pm2.19}$ | $24.12_{\pm1.45}$ | $14.00_{\pm1.00}$ | $9.04_{\pm1.48}$ | $26.66_{\pm2.29}$ | $16.00_{\pm2.00}$ |
| | DM(Zhao & Bilen, 2023) | $11.33_{\pm3.05}$ | $44.00_{\pm4.00}$ | $6.66_{\pm1.15}$ | $0.71_{\pm0.17}$ | $\mathbf{4.22}_{\pm0.32}$ | $151.66_{\pm2.51}$ | $7.30_{\pm1.09}$ | $33.65_{\pm1.98}$ | $11.33_{\pm0.57}$ | $\mathbf{10.95}_{\pm3.59}$ | $29.52_{\pm3.52}$ | $14.33_{\pm3.25}$ |
| | **Ours** | $\mathbf{27.33}_{\pm2.30}$ | $\mathbf{59.33}_{\pm7.02}$ | $\mathbf{3.83}_{\pm0.57}$ | $\mathbf{0.82}_{\pm0.10}$ | $4.16_{\pm0.40}$ | $\mathbf{143.00}_{\pm5.56}$ | $\mathbf{27.30}_{\pm1.98}$ | $\mathbf{42.54}_{\pm4.89}$ | $\mathbf{7.66}_{\pm1.15}$ | $6.43_{\pm3.11}$ | $\mathbf{34.52}_{\pm3.30}$ | $\mathbf{10.16}_{\pm1.60}$ |
| | Whole data | $45.33_{\pm5.03}$ | $76.00_{\pm2.00}$ | $1.83_{\pm0.28}$ | $2.11_{\pm0.42}$ | $8.59_{\pm0.60}$ | $15.34_{\pm0.46}$ | $43.49_{\pm2.39}$ | $54.91_{\pm6.34}$ | $3.66_{\pm2.08}$ | $17.14_{\pm0.71}$ | $44.76_{\pm1.79}$ | $7.16_{\pm0.288}$ |

**Audio-Visual Retrieval.** We have demonstrated that audio-visual distilled data facilitates learning effective audio-visual representations for event recognition. To further examine the audio-visual alignment, we explore whether distilled data could help learn a well-coordinated audio-visual space for cross-modal retrieval. Since our audio-visual distillation model focuses on semantic alignment rather than instance-level alignment, we evaluate audio-visual retrieval in a class-wise setting. Following (Gong et al., 2022), we create a retrieval test set by uniformly sampling a subset of five audio-visual samples per class from the original test split. We train the audio and visual ConvNet (Surís et al., 2018) with ArcFace margin loss (Deng et al., 2019). The shared classifier and margin loss help to learn a joint-modal embedding space with angular margins between classes. We train the model from scratch using the distilled data of IPC=20 and the same learning setting as the classification model.

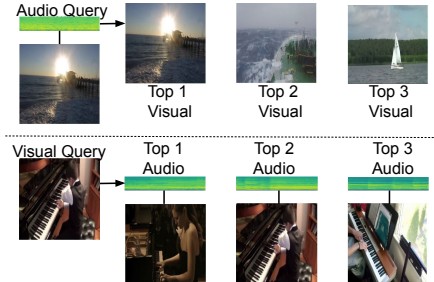

Figure 7: Audio-visual retrieval examples. We observe close alignment of audio queries with top visual results and vice versa.

We use these trained audio and visual components to get the corresponding representation of test samples and calculate the class retrieval recall at rank 1, 5, and median rank based on the cosine similarity. The results of audio-to-visual and visual-to-audio retrieval, in Tab. 6, demonstrate that our losses help distill audio-visual alignment (from real data) and hence our method outperforms DM in almost all scenarios. Fig. 7 visualizes the top-3 cross-modal retrieval from the VGG-10k test set. We observe that the retrieved modalities from the same class are closely aligned when trained with our audio-visual distilled data.

# 5 Conclusion and Discussion

In this paper, we explore a new task of multimodal distillation using audio-visual data. To evaluate the distilled data, we use audio-visual event recognition as the proxy task. To tackle this audio-visual dataset distillation problem, we first introduce a vanilla AVDM method that could learn audio- and visual-only distilled data and simply fuse the information for event recognition. We further improve the quality of condensed data by introducing two new losses to strengthen cross-modal alignment. We further show the improved alignment using the classwise cross-modal retrieval task. Experimental results on four audio-visual datasets show that our approach outperforms other methods consistently and audio-visual integration with condensed data is still helpful. This provides a new direction in the dataset distillation domain.

As the first attempt on this new task, our work has some limitations and introduces extents that are beyond the scope of a single paper but represent compelling directions for future research: (1) Compression-accuracy tradeoff: While our approach significantly reduces training data size, it does so at the cost of a performance drop (w.r.t whole data). Future work could explore methods to improve accuracy, potentially leveraging pre-trained weights (see Appendix F); (2) Short video segments: Our exploration focuses on short video segments. Extending our approach to long-form videos is a promising avenue for future research (see Appendix G); (3) Design for other audio-visual tasks: our approach learns synthetic data capturing class-level alignment between audio and visual modalities but lacks the instance-level detail needed for more complex audio-visual tasks. Future research will focus on creating synthetic data with improved instance-level alignment to enhance training for complex tasks such as audio-visual sound source localization. (4) Large datasets: Although our approach on VGGSound demonstrates better scalability and outperforms the baselines, the

overall performance remains limited. This limitation could be due to the large number of classes and the complex label space that needs to be learned from smaller datasets. This scenario is similar to ImageNet-1K for visual-only dataset distillation Zhou et al. (2022) performed poorly and required additional techniques like soft label alignment and reduced memory Cui et al. (2023) usage to enhance performance. These extensions are outside the scope of our current work, and we aim to tackle them in the future.

**Broader Impact Statement**

Given that audio-visual datasets may contain personally identifiable information, improper handling, storage, or sharing of distilled datasets could lead to privacy breaches and unauthorized access or misuse of sensitive data. Addressing these privacy concerns through the development of robust safeguards would be crucial.

**Acknowledgments**

This work was supported in part by an Amazon Research Award Fall 2023. Any opinions, findings, and conclusions or recommendations expressed in this material are those of the authors and do not reflect the views of Amazon.

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

# Appendix

In this supplementary, we first explain the direct cross-modal distribution losses that fail to align audio-visual synthetic data, our proposed algorithm, more details about herding and factor technique, additional implementation details, fusion methods, audio-visual retrieval, additional experiments and more visualizations.

## A    Failure Cross-Modal Distribution Matching

In this section, we delve into two direct cross-modal matching strategies that, contrary to intuition, do not enhance the alignment of audio-visual synthetic data within the audio-visual distribution matching framework. We explored the integration of alignment between audio and visual distilled data by experimenting with two simple cross-modal losses: Synthetic Cross-Modal Matching (SCMM) and Real-Synthetic Cross-Modal Matching (RSCMM).

**Synthetic Cross-Modal Matching(SCMM):** Given the natural synchronization in real audio-visual data it is logical to hypothesize that implementing a loss function to bring synthetic audio and visual distributions closer would improve the alignment of synthetic audio-visual data. Precisely, we show the SCMM loss in Eq. 11:

$$\mathcal{L}^{av}_{SCMM} = ||\frac{1}{|\mathcal{S}_a|}\sum_{j=1}^{|\mathcal{S}_a|}\psi_{\theta_a}(\mathcal{A}_\omega(\mathbf{s}^a_j)) - \frac{1}{|\mathcal{S}_v|}\sum_{j=1}^{|\mathcal{S}_v|}\psi_{\theta_v}(\mathcal{A}_\omega(\mathbf{s}^v_j))||^2 \tag{11}$$

**Real-Synthetic Cross-Modal Matching(RSCMM):** Another intuitive way to align synthetic audio-visual data with real audio-visual data is to directly match synthetic audio distribution with real visual data and vice-versa. Formally, we show the RSCMM loss in Eq. 14:

$$\mathcal{L}^{av}_{RS} = ||\frac{1}{|\mathcal{T}_a|}\sum_{i=1}^{|\mathcal{T}_a|}\psi_{\theta_a}(\mathcal{A}_\omega(\mathbf{x}^a_i)) - \frac{1}{|\mathcal{S}_v|}\sum_{j=1}^{|\mathcal{S}_v|}\psi_{\theta_v}(\mathcal{A}_\omega(\mathbf{s}^v_j))||^2 \tag{12}$$

$$\mathcal{L}^{va}_{RS} = ||\frac{1}{|\mathcal{T}_v|}\sum_{i=1}^{|\mathcal{T}_v|}\psi_{\theta_v}(\mathcal{A}_\omega(\mathbf{x}^v_i)) - \frac{1}{|\mathcal{S}_a|}\sum_{j=1}^{|\mathcal{S}_a|}\psi_{\theta_a}(\mathcal{A}_\omega(\mathbf{s}^a_j))||^2 \tag{13}$$

$$\mathcal{L}^{av}_{RSCMM} = \mathcal{L}^{va}_{RS} + \mathcal{L}^{av}_{RS} \tag{14}$$

We can further combine these losses with the vanilla audio-visual distribution loss ($\mathcal{L}^{av}_{base}$) to get a direct cross-modal matching loss, as shown in Eq. 15:

$$\mathcal{L}^{av}_{combined} = \mathcal{L}^{av}_{base} + \mathcal{L}^{av}_{SCMM} + \mathcal{L}^{av}_{RSCMM} \tag{15}$$

We do an ablation study to show the effect of different loss functions using one or more losses in Tab. 7. We observe a negative effect of adding these direct cross-modal losses and observe a severe drop in accuracy. This could be explained by the modality gap that is created by using randomly initialized feature extractors ($\psi_a, \psi_v$). To visualize the effect of these losses we plot the distribution of the first three classes of VGGS-10k data (as shown in Fig. 8) and observe that these losses lead to unstable training with the synthetic data unable to cover the real audio-visual distribution.

Table 7: Ablation study on the different loss terms and different datasets (IPC=10). Here *Base*, SCMM, and RSCMM are vanilla audio-visual distribution matching, Synthetic Cross-Modal Matching, and Real-Synthetic Cross-Modal Matching respectively. We observe that adding SCMM and RSCMM to the baseloss results in lower performance.

| Random | Herding | Factor | Base | SCMM | RSCMM | VGGS-10k | AVE |
|--------|---------|--------|------|------|-------|----------|-----|
| ✓ | | | | | | $32.01_{\pm1.64}$ | $20.00_{\pm1.45}$ |
| | ✓ | | | | | $39.89_{\pm1.64}$ | $26.86_{\pm0.52}$ |
| | ✓ | ✓ | | | | $40.28_{\pm2.34}$ | $31.80_{\pm1.28}$ |
| | ✓ | ✓ | ✓ | | | $\mathbf{45.31}_{\pm2.68}$ | $\mathbf{34.80}_{\pm1.68}$ |
| | ✓ | ✓ | ✓ | ✓ | | $31.15_{\pm2.87}$ | $29.55_{\pm1.36}$ |
| | ✓ | ✓ | ✓ | | ✓ | $30.22_{\pm3.19}$ | $29.12_{\pm1.29}$ |
| | ✓ | ✓ | ✓ | ✓ | ✓ | $24.38_{\pm2.21}$ | $26.13_{\pm0.97}$ |

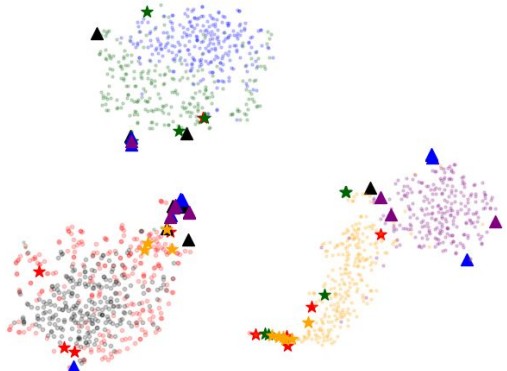

Figure 8: Distribution of synthetic audio-visual data (IPC=10 and $l$=1) generated by using the direct cross-modal matching approach(i.e. SCMM and RSCMM). (green,blue), (red,**black**) and (purple,yellow) points are the real (audio, visual) points for the first 3 classes of VGGS-10k. The synthetic (audio, visual) data is represented by (★, ▲). Note that the model trained on whole VGGS-10k data is used for extracting the embeddings. We observe significant misalignment of the synthetic audio-visual data, which can be attributed to unstable training losses.

# B Herding and Factor technique

Herding iteratively selects data points (features) that are closest to the cumulative mean of previously selected points, aiming to create a subset that closely represents the overall data distribution. We use the pseudocode in Fig. 9 to get the data subset of IPC size for each class.

```
mean = torch.mean(features, dim=0, keepdim=True)
idx_selected = []
idx_left = np.arange(features.shape[0]).tolist()
for i in range(IPC):
    det = mean*(i+1) - torch.sum(features[idx_selected],dim=0)
    dis = distance(det, features[idx_left])
    idx = torch.argmin(dis)
    idx_selected.append(idx_left[idx])
    del idx_left[idx]
```

Figure 9: Pseudocode for Herding.

The factor technique (Zhang et al., 2023; Kim et al., 2022; Zhao et al., 2023) aims to increase the number of representations extracted from $S$ without additional storage cost. Specifically, with the factor parameter

---

**Algorithm 1** Audio-Visual Dataset Distillation

---

**Require:** Initial set of synthetic samples $S$ for $C$ classes, where each class $c$ is represented by a subset $S_c$ of synthetic data , deep neural network $\psi_{\theta_a}$ parameterized with $\theta_a$ and probability distribution over $P_{\theta_a}$, $\psi_{\theta_v}$ parameterized with $\theta_v$ and probability distribution over $P_{\theta_v}$, differentiable augmentation $A_\omega$ parameterized with $\omega$, augmentation parameter distribution $\Omega$, training iterations $K$, learning rate $\eta$, loss weights $\lambda_{ICM}$, $\lambda_{CGM}$.
**Input:** Real training set $T$
**for** $k = 0, \ldots, K-1$ **do**
    Sample $\theta_a \sim P_{\theta_a}$, $\theta_v \sim P_{\theta_v}$
    Sample mini-batch pairs $B_c^T \sim T$ and $B_c^S \sim S$ and $\omega_c \sim \Omega$ for every class $c$
    **for** $c = 0, \ldots, C-1$ **do**
        $\bar{R}_a^c = \frac{1}{|B_c^T|} \sum_{(x^{av},y) \in B_c^T} \psi_{\theta_a}(A_{\omega_c}(x^a))$                  ▷ Audio real mean embedding
        $\bar{R}_v^c = \frac{1}{|B_c^T|} \sum_{(x^{av},y) \in B_c^T} \psi_{\theta_v}(A_{\omega_c}(x^v))$                  ▷ Visual real mean embedding
        $\bar{S}_a^c = \frac{1}{|B_c^S|} \sum_{(x^{av},y) \in B_c^S} \psi_{\theta_a}(A_{\omega_c}(x^a))$               ▷ Audio synthetic mean embedding
        $\bar{S}_v^c = \frac{1}{|B_c^S|} \sum_{(x^{av},y) \in B_c^S} \psi_{\theta_v}(A_{\omega_c}(x^v))$               ▷ Visual synthetic mean embedding
        $\mathcal{L}_{base} = ||\bar{R}_a^c - \bar{S}_a^c||^2 + ||\bar{R}_v^c - \bar{S}_v^c||^2$
        $\mathcal{L}_{ICM} = ||(\bar{R}_a^c + \bar{R}_v^c) - (\bar{S}_a^c + \bar{S}_v^c)||^2$
        $\mathcal{L}_{CGM} = ||(\bar{R}_a^c + \bar{S}_v^c) - (\bar{R}_v^c + \bar{S}_a^c)||^2$
        Compute $\mathcal{L} = \mathcal{L}_{base} + \lambda_{ICM} \cdot \mathcal{L}_{ICM} + \lambda_{CGM} \cdot \mathcal{L}_{CGM}$
        Update $S_c \leftarrow S_c - \eta \nabla_S \mathcal{L}$
    **end for**
**end for**
**Output:** $S$

---

being $l$, each image $s_i^{av} \in S$ is factorized into $l \times l$ mini-examples and then up-sampled to its original size in training. We show an example for audio datapoint but it can be similarly extended to visual datapoint.

$$s_{a_i} \xrightarrow[\text{Factor}]{} \begin{bmatrix} s_{a_i}^{1,1} & \cdots & s_{a_i}^{1,l} \\ \vdots & \ddots & \vdots \\ s_{a_i}^{l,1} & \cdots & s_{a_i}^{l,l} \end{bmatrix} \xrightarrow[\text{Up-sample}]{} \left\{ s_{a_i}^{'1}, s_{a_i}^{'2}, \ldots, s_{a_i}^{'l \times l} \right\}.$$

We conduct an ablation study to find the optimal value of $l$ in Tab. 8 and observe that $l = 2$ gives the best performance for VGGS-10k at IPC=10. The performance of $l = 3$ is slightly worse as it discards too many details in each partition.

Table 8: Ablation study of the number of partition $l^2$ in our augmentation on VGGS-10k with 10 Img/Cls.

| Partition $(l \times l)$ | $1 \times 1$ | $2 \times 2$ | $3 \times 3$ |
|---|---|---|---|
| Accuracy | $45.66_{\pm 2.53}$ | $54.99_{\pm 1.73}$ | $54.53_{\pm 1.57}$ |

To further demonstrate the advantage of our method, we provide the test accuracy with training steps in Fig. 10. We observe that our method out-perform DM at different factor $l$ values and herding-based initialization improves the initial performance.

## C  Detailed model architecture and hyperparameters

Following previous distillation methods (Zhao et al., 2023; Zhao & Bilen, 2023; Cazenavette et al., 2022), we use the ConvNet architecture model (Zhao et al., 2020) for both audio and visual data inputs. The audio ConvNet consists of 3 blocks with each block consisting of a 128-kernel convolution layer, instance normalization, ReLU, and average pooling layer. The last average pooling layer is replaced by an adaptive average pooling layer with (7,7) spatial filter to match the dimension of visual embedding (of size 6272) to facilitate joint matching. For the large image input, we use 5 such blocks. We use the same audio and visual model architecture for all the experiments.

For our approach, we use a learning rate of 0.2 for audio and image synthetic data and SGD optimizer with a momentum of 0.5. We initialize our synthetic data with Herding-selected AV pairs(elaborated in the coreset selection section) and use a real audio-visual pair batch size of 128 at each iteration. For images per

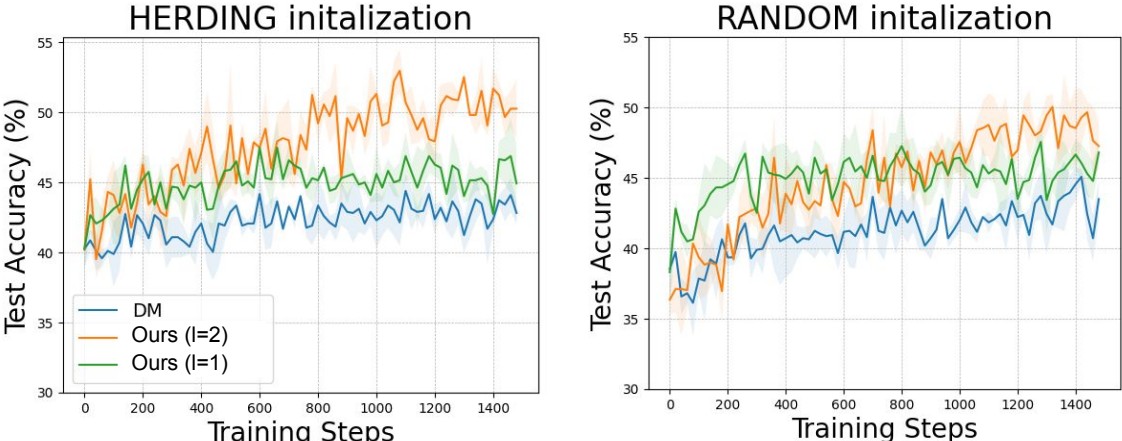

Figure 10: Performance comparison between Distribution Matching(DM) and our approach with different factor ($l$) values, across varying training steps for VGGS-10k dataset with IPC=10. The left and right plots differ in initialization i.e. herding-based and random initialization respectively. We observe that herding initialization gives a better initialization performance and our approach performs better than DM at different factor values.

class(IPC) 1, and 10 we keep $\lambda_{ICM}=\lambda_{CGM}=10$, while for IPC 20 we keep $\lambda_{ICM}=\lambda_{CGM}=20$. Following the previous works(Zhang et al., 2023; Kim et al., 2022; Zhao et al., 2023) we keep the factor parameter $l$ to 2. We sample audio at 11kHz and transform them into log mel-spectrograms using a hop length of 200 and 128 mel banks, resulting in 128x56 size. The whole dataset indicates training on the entire real training set.

During evaluation, the initial learning rate for the audio model is kept 1e-3, the visual part is kept 1e-4 and for the classifier layers are kept at 1e-4. The learning rates are lowered by multiplying by 0.1 after every 10 epochs.

## C.1 Baseline implementations

**Trajectory Matching(MTT).** Contrary to DM, MTT involves a range of hyperparameters that require optimization. For MTT, we retain 20 expert trajectories for each modality independently and conduct a hyperparameter search (memory-constrained) across all synthetic images per class (IPC) and dataset configurations. For the Audio-visual MTT, we separately train distilled data for audio-only and visual-only MTT and then combine these datasets using an ensemble fusion to obtain AV results. The best hyperparameters for each configuration are detailed in Tab. 9.

**DC and DSA.** The original methods are proposed for image-only dataset distillation. For audio-visual datatset distillation, we extend the method to independently learn synthetic data for both modalities. Further we use this distilled data for evaluation. We follow the optimal hyperparameters suggested by the authors (Zhao et al., 2020; Zhao & Bilen, 2021) and due to large image resolution, it does not scale to visual-only distillation.

# D Audio-Visual Fusion

Audio-visual fusion is critical for multimodal model performance. We examined how different fusion strategies affect models trained on distilled synthetic data. We compared several fusion approaches: Sum, Concatenation (Concat), Ensemble, and audio-guided visual attention (Attention) (Pian et al., 2023). The fusion function takes in audio feature $f^a$ and visual feature $f^v$ to get fused feature $f^{av}$. Here are the formulations of the fusion functions:
**Sum:** It directly sums the audio and visual modality features: $f^{av} = f^a + f^v$.

Table 9: Hyper-parameters for best performing Trajectory Matching(MTT) distillation experiments, for all the experiments learning rate of step size is kept $10^{-5}$. "-" denotes the configuration for which the method could not scale up.

| Modlity | Dataset | IPC | Synthetic Steps | Expert epochs | Max Start Epoch | Synthetic Batch Size | Learning rate (Pixels) | Starting synthetic step size |
|---------|---------|-----|-----------------|---------------|-----------------|----------------------|------------------------|------------------------------|
| A-only | VGGS-10K | 1 | 10 | 2 | 2 | 20 | $10^3$ | $10^{-3}$ |
| | | 10 | 10 | 2 | 2 | 20 | $10^3$ | $10^{-3}$ |
| | | 20 | 10 | 2 | 2 | 10 | $10^4$ | $10^{-3}$ |
| | VGGSound | 1 | 10 | 2 | 2 | 20 | $10^3$ | $10^{-3}$ |
| | | 10 | 10 | 2 | 2 | 20 | $10^3$ | $10^{-3}$ |
| | | 20 | 5 | 2 | 2 | 2 | $10^1$ | $10^{-3}$ |
| | MUSIC-21 | 1 | 10 | 2 | 2 | 20 | $10^3$ | $10^{-3}$ |
| | | 10 | 10 | 2 | 2 | 20 | $10^3$ | $10^{-3}$ |
| | | 20 | 5 | 2 | 2 | 2 | $10^1$ | $10^{-3}$ |
| | AVE | 1 | 20 | 2 | 2 | 20 | $10^3$ | $10^{-3}$ |
| | | 10 | 10 | 2 | 2 | 20 | $10^3$ | $10^{-3}$ |
| | | 20 | 10 | 2 | 2 | 10 | $10^4$ | $10^{-3}$ |
| V-only | VGGS-10K | 1 | 5 | 2 | 2 | 5 | $10^3$ | $10^{-4}$ |
| | | 10 | 5 | 2 | 2 | 5 | $10^4$ | $10^{-4}$ |
| | | 20 | 10 | 2 | 2 | 5 | $10^1$ | $10^{-4}$ |
| | VGGSound | 1 | 5 | 2 | 2 | 5 | $10^3$ | $10^{-4}$ |
| | | 10 | - | - | - | - | - | - |
| | | 20 | - | - | - | - | - | - |
| | MUSIC-21 | 1 | 5 | 2 | 2 | 5 | $10^3$ | $10^{-4}$ |
| | | 10 | 5 | 2 | 2 | 5 | $10^3$ | $10^{-4}$ |
| | | 20 | - | - | - | - | - | - |
| | AVE | 1 | 5 | 2 | 2 | 5 | $10^3$ | $10^{-4}$ |
| | | 10 | 2 | 2 | 2 | 1 | $10^5$ | $10^{-4}$ |
| | | 20 | - | - | - | - | - | - |

**Concatenation:** It directly concatenates the audio and visual modality features: $f^{av} = [f^a; f^v]$.

**Ensemble:** To get the ensemble prediction $p_{av}$ the predictions from the audio and visual modality features are averaged. Given the audio and visual learnable projection matrices $W^a$ and $W^v$ respectively, we can define $p_{av}$ as

$$p^a = Softmax(f^a W^a)$$
$$p^v = Softmax(f^v W^v)$$
$$p^{av} = \frac{p^a + p^v}{2}$$

**Attention:** Audio-guided visual attention has shown to be an effective mechanism to learn correlations between audio and visual features adaptively (Pian et al., 2023; Li et al., 2021; Tian et al., 2018). Given learnable projection matrices $W^a, W^v, U^a, U^v$ and $\odot$ as the Hadamard product, we can formally define attention fusion $f^{av}$ as:

$$\text{Score}^a = \tanh(f^a W^a),$$
$$\text{Score}^v = \tanh(f^v W^v),$$
$$w = Softmax(\text{Score}^a \odot \text{Score}^v)$$
$$f'^v = f^v \odot w$$
$$f^{av} = \tanh(f^a U^a) + \tanh(f'^v U^v)$$

## D.1 Why does attention fusion perform worst?

We note three reasons for this.

**Classwise alignment.** Our three losses aim at learning a classwise distribution matching (vanilla audio-visual DM loss) and classwise alignment matching (Joint Matching and Modality Gap Matching loss). Our distilled audio-visual data does not have any instance-wise constraints (we observed some performance degradation on adding such constraints) and hence training an attention-based fusion which relies on the paired audio-visual correspondence between the training examples, falls short in performance.

**Spatial Distortion.** Since all the image and audio-spectrogram pixel values are learnable parameters and no additional constraint is added to preserve the spatial information from the original initialized data, the distilled audio-visual data loses spatial information like object boundaries and frequency boundaries. Hence while training, the distilled audio gets confused about which part of the distilled image to look at.

**Number of parameters.** In addition to the classwise alignment and spatial distortion, we observe that the comparatively large number of model parameters (shown in Tab. 10) and the small condensed data add another hindrance to audio-guided attention performance.

Table 10: Comparison of Learnable Parameters

| Fusion | Parameters |
|---|---|
| Attention | 192,798 |
| Concatenation | 125,450 |
| Sum | 62,730 |
| Ensemble | 125,360 |

# E   Audio-visual retrieval

The framework for training Audio-visual retrieval using distilled data is shown in Fig. 11. We alternatively train audio and visual encoder with ArcFace loss (Deng et al., 2019). The loss can be formulated as in eq. 16. Where $\theta$ is the angle between the audio (or visual) embeddings and mean class embeddings, m is the additive angular margin penalty and s is the scaling factor. For our case, we keep s = 3.0 and m=0.2.

$$L = -\log \frac{e^{s\cos(\theta_{y_i}+m)}}{e^{s\cos(\theta_{y_i}+m)} + \sum_{j=1,j\neq y_i}^{N} e^{s\cos\theta_j}} \tag{16}$$

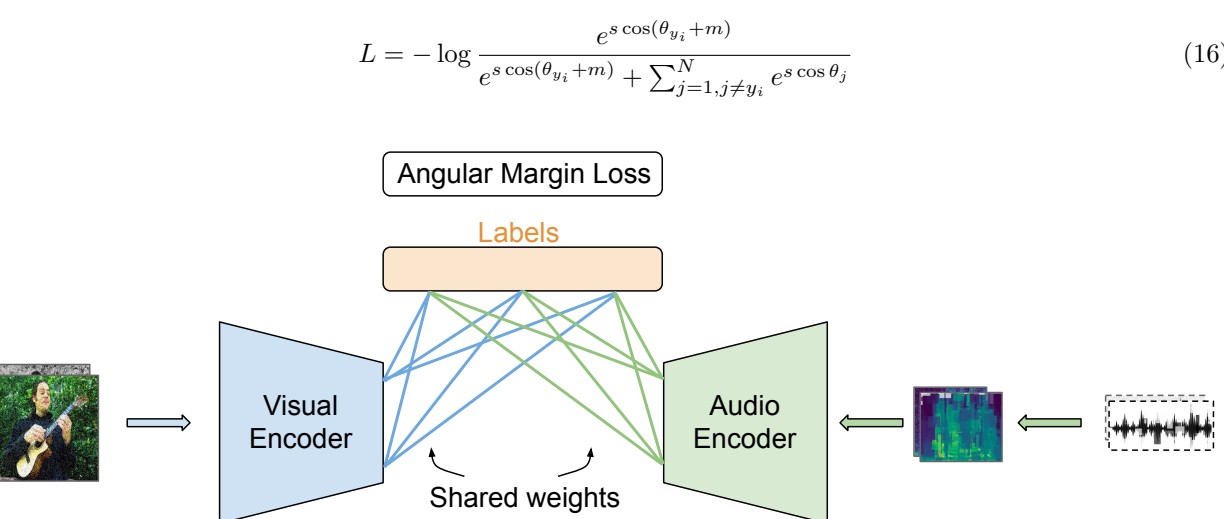

Figure 11: Framework for Audio-visual retrieval using distilled data. We have shared classifier weights and we use arcface margin loss to get discriminative class embeddings.

## E.1   Other details

**VGGS-10k categories.** We create a subset of VGGSound(Chen et al., 2020c) for initial experiments and analysis. The selected categories are playing piano, playing acoustic guitar, police radio chatter, toilet flushing, driving buses, chicken crowing, child speech or kid speaking, basketball bounce, fireworks banging, and ocean burbling.

**Compute resources.** We test our experiments on A5000 and A6000 GPUs with 24 GB memory and 48 GB memory respectively.

**Ablation study of loss weights** $\lambda_{ICM}$, $\lambda_{CGM}$. We do an ablation study over the effect of loss weights $\lambda_{ICM}$, $\lambda_{CGM}$ for different IPCs for VGGS-10k dataset and show the results in Tab. 11. From the ablation, we observe that the highest accuracy for IPC 1, 10, 20 occur at $\lambda_{ICM} = \lambda_{CGM} = 10, 20, 20$ respectively. Subsequently, we choose $\lambda_{ICM} = \lambda_{CGM} = 10$ for IPC 1 and $\lambda_{ICM} = \lambda_{CGM} = 20$ for IPC 20 for all other datasets. However, we observe $\lambda_{ICM} = \lambda_{CGM} = 10$ working better for IPC 10 for other datasets (especially bigger datasets) and hence we keep it 10 for IPC 10. Due to limited computational resources, we do not show ablation over other datasets.

Table 11: Ablation study of the loss weights $\lambda_{ICM}$, $\lambda_{CGM}$ for VGGS-10k dataset with different IPC setting. Here $\lambda_{ICM} = \lambda_{CGM} = \lambda$

|     |    | $\lambda$ | | |
|-----|----|------|------|------|
|     |    | 1 | 10 | 20 |
| IPC | 1  | $39.07_{\pm 2.98}$ | $40.41_{\pm 1.81}$ | $39.62_{\pm 2.03}$ |
|     | 10 | $48.92_{\pm 2.43}$ | $54.99_{\pm 1.73}$ | $56.19_{\pm 1.62}$ |
|     | 20 | $49.48_{\pm 1.68}$ | $56.99_{\pm 0.80}$ | $58.04_{\pm 1.68}$ |

# F    Pretrained initialization

We show that pretraining can help improve the performance of our approach and can further close the gap with the whole-data. In Tab. 12 we observe that initializing evaluation model with VGGSound trained weights can reduce the gap with whole-data from ∼12% to ∼3% for AVE data.

Table 12: Comparison of different initilization for distilled data at IPC=20. Here Pretrained initialization is of model initialized with VGGSound trained models.

|             | Random init. | Pretrained init. | Whole data |
|-------------|--------------|------------------|------------|
| AVE(IPC=20) | $40.13_{\pm 1.00}$ | $48.71_{\pm 1.31}$ | $52.20_{\pm 0.38}$ |

# G    Extending to 10-sec videos

In Tab. 13 we extend our problem setting to 10 secs videos. We observe that the trends for 10-sec are similar to 1-sec problem setting.

Table 13:   Performance comparison for 10 sec video segments

|          | IPC | Random | Herding | DM | Ours |
|----------|-----|--------|---------|-----|------|
| VGGS-10K | 1   | $25.33_{\pm 2.01}$ | $22.21_{\pm 0.78}$ | $25.33_{\pm 2.01}$ | $\mathbf{39.14}_{\pm 3.59}$ |
|          | 20  | $46.10_{\pm 1.97}$ | $49.25_{\pm 1.26}$ | $49.75_{\pm 2.14}$ | $\mathbf{53.05}_{\pm 2.36}$ |
| AVE      | 1   | $9.50_{\pm 1.22}$ | $12.84_{\pm 0.70}$ | $12.54_{\pm 0.92}$ | $\mathbf{17.22}_{\pm 2.52}$ |
|          | 20  | $27.36_{\pm 2.01}$ | $30.00_{\pm 0.98}$ | $30.05_{\pm 2.34}$ | $\mathbf{44.98}_{\pm 1.04}$ |

# H    Additional Visualizations

Figures 12 and 13 visualize the synthetic data for VGGS-10k for IPC=1 with $l = 1$ and $l = 2$, respectively. We observe that all the audio-visual distilled data consists of repeating patterns and is more prominent in the visual data. Due to the repetitive patterns in Fig. 12 ($l = 1$) we can ignore the fine-grained details and achieve better performance by utilizing the storage (as in Fig. 13 ($l = 2$)) without increasing the synthetic size. We also observe similar patterns in Fig. 14 when trained with DM. We further show that the data becomes less far away from the original initialized real data in Figures 15, where we visualize VGGS-10k with IPC=10. Fig. 16,17,18 visualizes the audio-visual data learnt through our method for IPC=1 and VGG, Music-21 and AVE datasets, respectively.

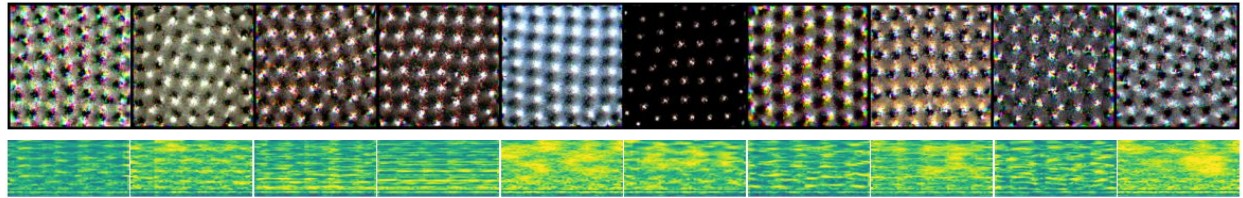

Figure 12: Visualization of audio-visual synthetic data for VGGS-10k for IPC=1, learned through our proposed Audio-Visual Data Distillation approach ($l = 1$).

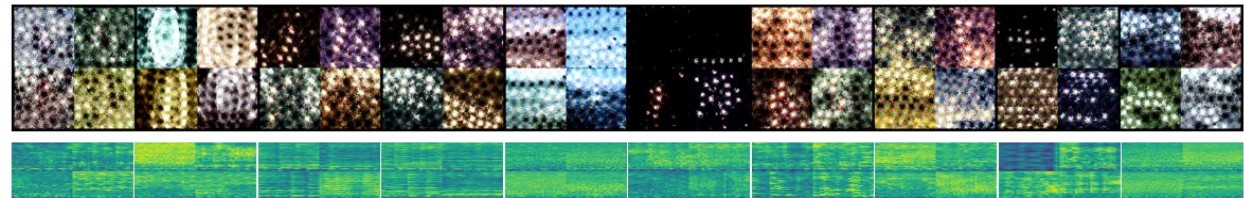

Figure 13: Visualization of audio-visual synthetic data for VGGS-10k for IPC=1, learned through our proposed Audio-Visual Data Distillation approach ($l = 2$).

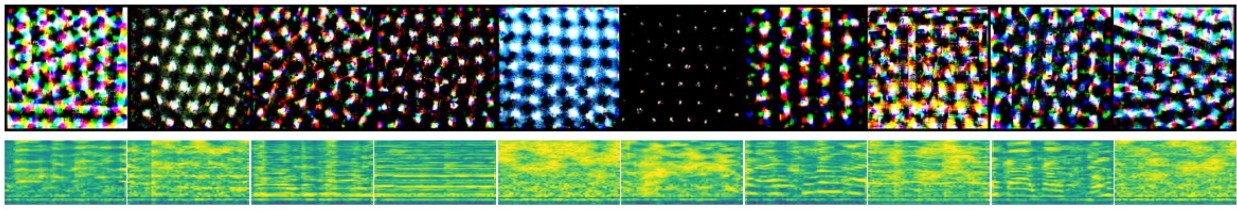

Figure 14: Visualization of audio-visual synthetic data for VGGS-10k for IPC=1, learned through DM method.

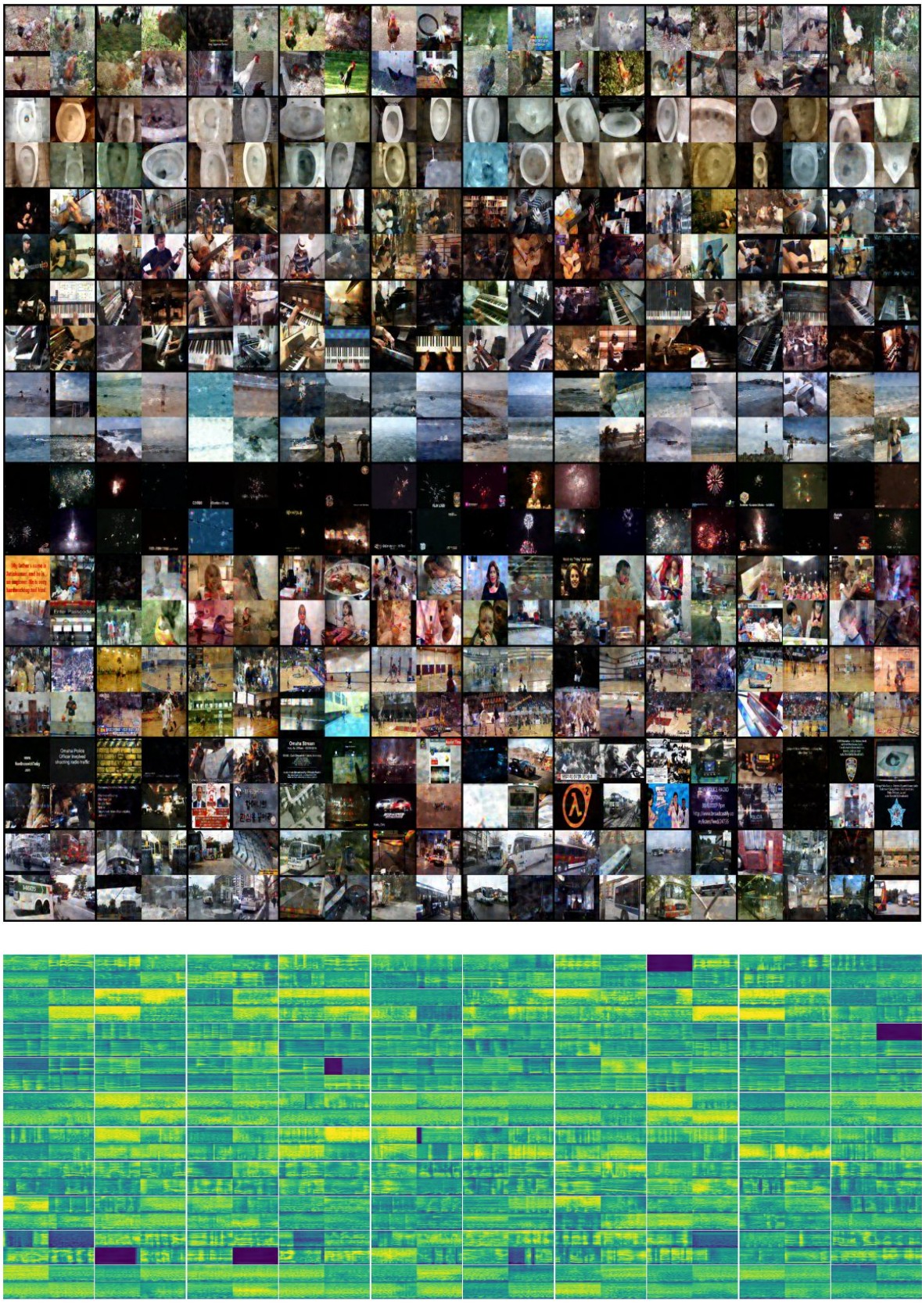

Figure 15: Visualization of audio-visual synthetic data for VGGS-10k for IPC=10, learned through our proposed Audio-Visual Data Distillation approach

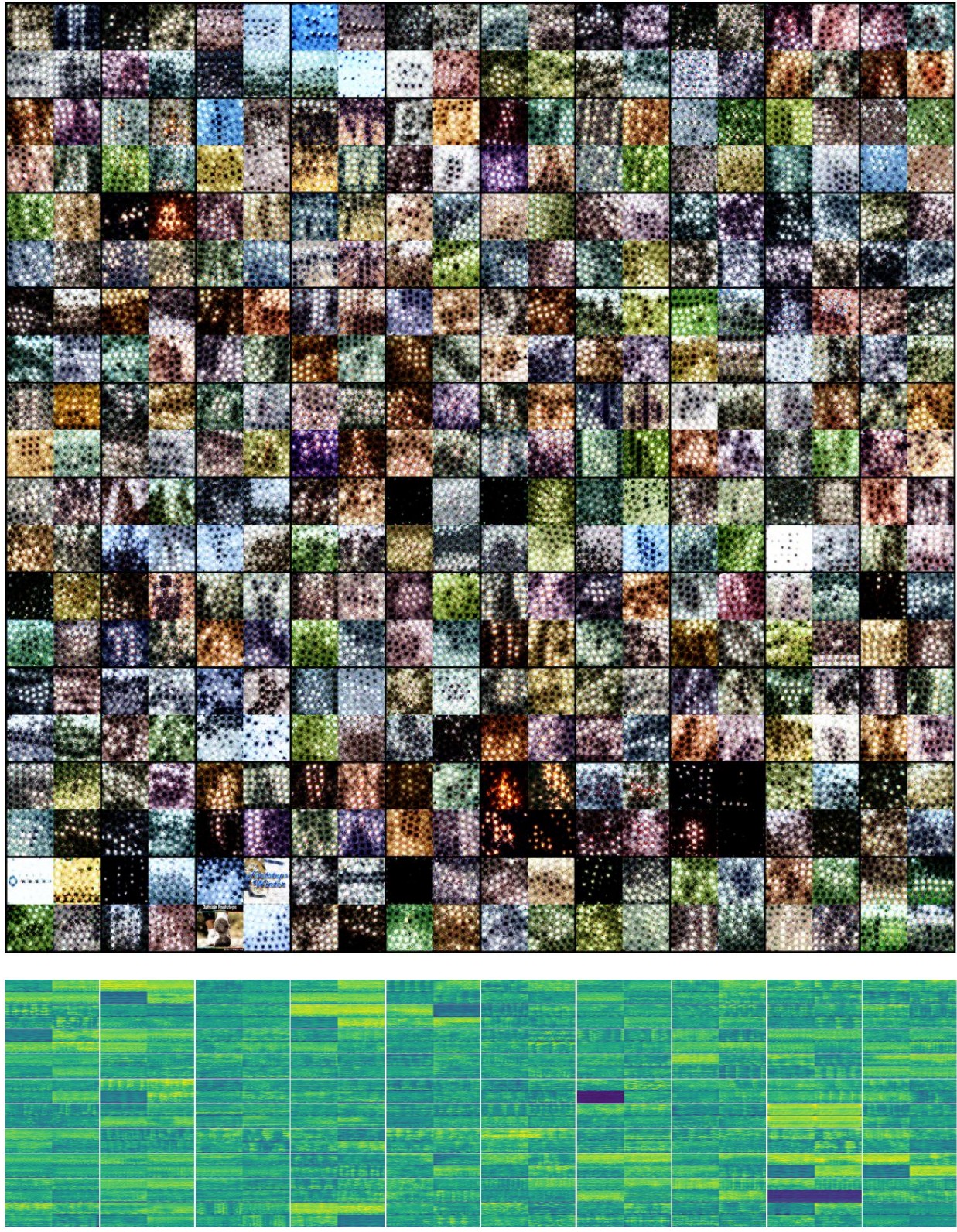

Figure 16: Visualization of audio-visual synthetic data for first 100 classes of VGGSound for IPC=1, learned through our proposed Audio-Visual Data Distillation approach

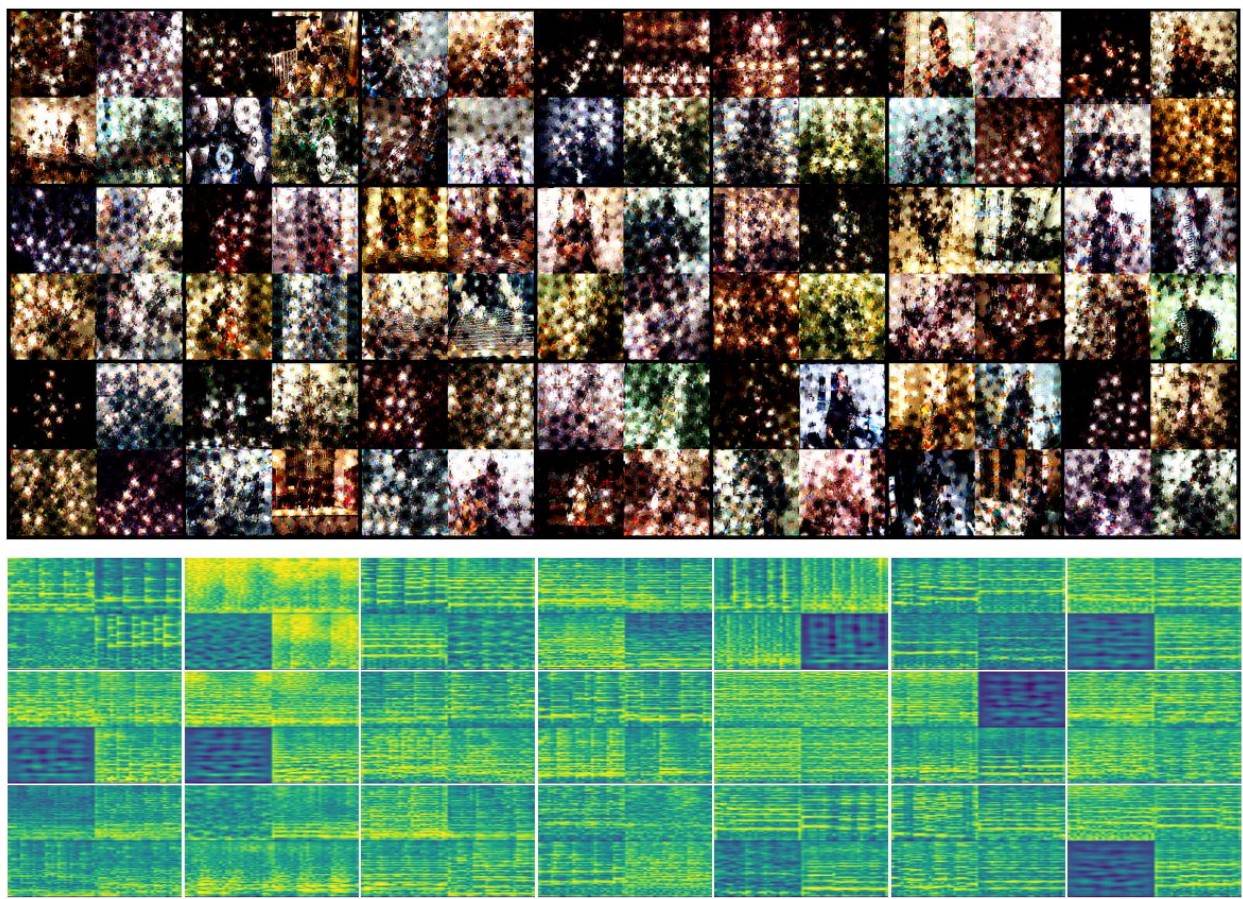

Figure 17: Visualization of audio-visual synthetic data for Music-21 for IPC=1, learned through our proposed Audio-Visual Data Distillation approach

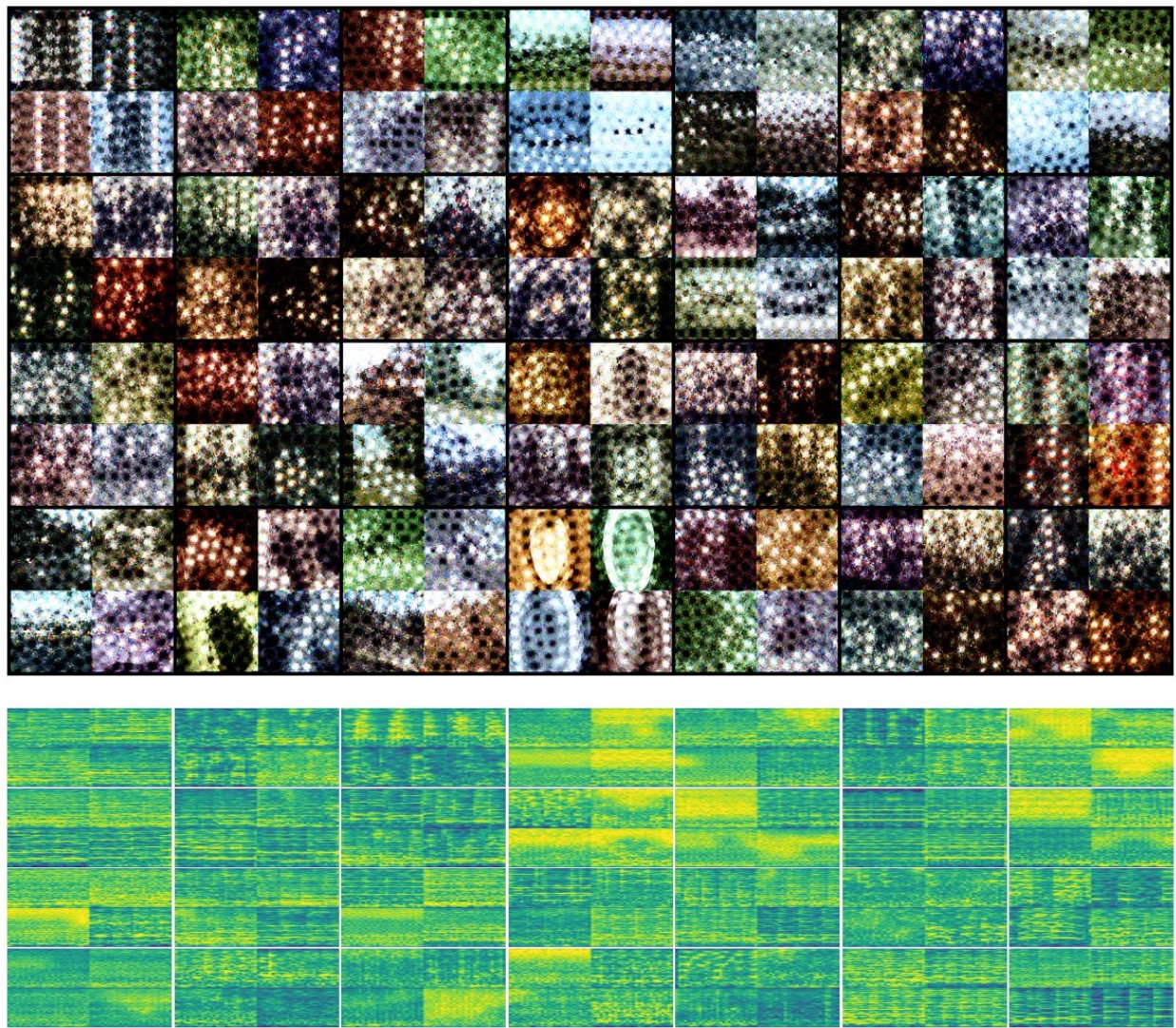

Figure 18: Visualization of audio-visual synthetic data for AVE for IPC=1, learned through our proposed Audio-Visual Data Distillation approach

