# OpenReview forum: "Audio-Visual Dataset Distillation"
_TMLR — Accepted by TMLR_

### Review · Reviewer_s2fd · 2024-08-02

**Summary Of Contributions:**

This article introduces a new audio-visual dataset distillation task to construct a condensed and representative synthetic audio-visual dataset that maintains the cross-modal semantic association. To achieve this task, the article introduces an audio-visual distribution matching framework and investigates the effectiveness of audio-visual integration and various multi-modal fusion methods on four datasets. The authors provide thorough experimental validation, demonstrating the superiority of their method over existing state-of-the-art distillation methods. The extensive experiments underline the effectiveness of the proposed framework.

**Audience:**

Yes

**Claims And Evidence:**

Yes

**Requested Changes:**

1. It is suggested to indicate the trainable model modules in Figure 3 or caption. Besides, the initialization of synthetic data is to select a subset of the real data. But why the initialization of synthetic data in Figure 3 is blurred and different from the real images? It should be better to draw the initialized synthetic data separately from the updated synthetic data during the training process, which can better illustrate the initial state and the intermediate updated state of the synthetic data.

2. The naming of the loss is not intuitive enough. "Joint matching" and "modality gap matching" do not adequately describe the relationships between Rv, Ra, Sv, and Sa. If the names could emphasize “cross modalities” or “cross data spaces”, it would be clearer.

3. The article is somewhat redundant, repeatedly using phrases like "align the modality gap between real and synthetic data" and "enforces cross-modal matching between real and synthetic data."

4. Figure 6 demonstrates that the proposed method can better capture the underlying distribution of the real data. However, both the herding and the MMD loss are based on the selection and calculation of the mean center of the real distribution. Is there a rational explanation for why the proposed method can obtain more dispersed distillation data?

5. Some typos: the font for L in Algorithm 1 is incorrect; the Tab index in Appendix E is not displayed.

**Strengths And Weaknesses:**

Strengths:

1. The motivation is clear, and the problem itself is important for the community.

2. The proposed method is easy to follow.

Weaknesses:

1. The writing of this paper should be improved.

2. Some analytic experiments should be added. See the requested changes below.

---

> ### Author Response · Authors · 2024-08-08
> **Response to the requested changes**
>
> We sincerely thank the Reviewer for the time and effort in reviewing the paper and providing valuable comments! As per the reviewer's suggestions, we have revised our manuscript, and the updated parts are highlighted using blue color. We are happy to make any further changes as requested.
>
> 1. Figure 3 and Synthetic Data Initialization.
>
>      - (Response): Thank you for the suggestion! (1) We have updated Figure 3 to show the learnable components by adding backpropagation to synthetic audio and visual data. (2) You're right, synthetic data is initialized using real data, and the figure illustrates an intermediate stage of training, resulting in slightly blurred images. We've updated the caption to clarify this. While showing both initial and intermediate states of distilled data might be helpful, including them in the same figure would compromise its clarity and space. However, we illustrate this in Figure 5, showcasing how initial real data points are modified to become the final synthetic data.
>
>
> 2. The naming of the loss is not intuitive enough.
>
>     - (Response):  Nice suggestion! We agree and have updated "Joint matching(JM)" -> "Implicit cross-matching (ICM)" and "modality gap matching(MGM)" -> "cross-modal gap matching (CGM)" in our manuscript.
>
> 3. Redundant phrases.
>
>     - (Response):  We have updated our manuscript to remove the redundant usage of "alignment" and "cross-modal matching" phrases.
>
> 4. Why can the proposed method obtain more dispersed distillation data?
>     - (Response): We observed that DM and MTT tend to cluster distilled data closely around the modality mean centers, while our method generates more dispersed data. This dispersion is due to the additional cross-modal matching losses we incorporate, which act as a form of regularization. Unlike DM and MTT, which have independent modality-wise matching losses, our method encourages the distilled data to better reflect the true underlying distribution of real audio and visual data.
>
> 5. Some typos: the font for L in Algorithm 1 is incorrect; the Tab index in Appendix E is not displayed.
>     - (Response): Thanks for the suggestion! We have fixed these typos.
>
> Thank you again for your valuable feedback. We welcome any further questions or suggestions you may have.

---

### Review · Reviewer_CF7n · 2024-08-20

**Summary Of Contributions:**

This paper mainly progresses and improves the multimodal (audio-visual) dataset distillation task. The author put forward the DM methods tailored for multimodal datasets, which have been summarized into three specific loss functions: Vanilla Audio-Visual Distribution Matching loss, Implicit Cross-Matching loss, and Cross-Modal Gap-Matching loss. These contributions have been verified through two distinct tasks (Audio-visual event recognition and Audio-visual retrieval), demonstrating the robustness and effectiveness of their approach.

**Audience:**

Yes

**Broader Impact Concerns:**

The DD method helps protect the privacy information in the original dataset and the Broader Impact Statement in the submission paper is sufficient enough.

**Claims And Evidence:**

Yes

**Requested Changes:**

Critical:
- The main experimental results (Table 3) are obtained with the help of the Herding and Factor technique. And there are no ablations regarding these two technologies. Even in Appendix A (Table 7), the best results are obtained with the base loss and the above two technologies. There lack of sufficient experimental evidence on whether the progress of Acc comes from the proposed DM methods or the high-quality initialization technique.
- The description of “upper bound” used in the contents and tables is inappropriate, as literature[1,2] has proved that the Acc is beyond the whole dataset just using the distilled dataset.

Strengthen:
- In Algorithm 1, the Update process, the subscript of the updated dataset $\mathcal{S}_{c}$ should be $c+1$.
- The failure of MTT on some datasets should be explained more clearly. Is it invalid with the OOM error or not suitable for the task?
- In Figure 3, the discrepancy gap between the original and distilled pair is different from the visualization results in Figure 5, the author should modify the caption of Figure 3, especially for the last sentence.

Minor:
- The latex codes within tables are different.

[1] Towards Lossless Dataset Distillation via Difficulty-Aligned Trajectory Matching. ICLR2024.

[2] Prioritize Alignment in Dataset Distillation.

**Strengths And Weaknesses:**

Strengths:
- The theoretical and empirical analysis supports the construction of the proposed DM-based multimodal DD methods.
- The well-defined loss functions with demonstrated interpretability and robustness through extensive experiments.

Weaknesses:
- Some main experiments to verify the effectiveness of the proposed matching losses are missing.
- See “Requested Changes” for more details.

---

> ### Author Response · Authors · 2024-08-28
> **Response to the requested changes**
>
> We sincerely thank the Reviewer for the time and effort in reviewing the paper and providing valuable comments! As per the reviewer's suggestions, we have revised our manuscript, and the updated parts are highlighted using blue color. We are happy to make any further changes as requested.
>
> Critical:
>
> 1. Additional ablations to evaluate accuracy contributions:
>    - (Response) Thank you for the suggestion. We did additional ablation studies to evaluate the accuracy obtained from various combinations of herding and factor along with our proposed matching losses (ICM & CGM).
>    - Our results show that the proposed matching losses enhance performance over the base loss, and this improvement is further amplified when combined with the herding and(or) factor method.
>    - Hence, all the additional losses positively contribute to the overall accuracy.
>    - We have further added these ablation experiments in Table 4 of the main paper.
>
> | Random | Herding | Factor | Base | ICM + CGM | VGGS-10k       | AVE            |
> |:------:|:-------:|:------:|:----:|:---------:|:--------------:|:--------------:|
> |   ✓    |         |        |  ✓   |           | 43.85 ± 1.75   | 28.14 ± 1.80   |
> |   ✓    |         |        |  ✓   |     ✓     | 45.87 ± 2.11   | 30.71 ± 1.17   |
> |        |    ✓    |        |  ✓   |           | 44.47 ± 1.96   | 32.36 ± 0.72   |
> |        |    ✓    |        |  ✓   |     ✓     | 46.68 ± 2.27   | 33.60 ± 0.66   |
> |   ✓    |         |    ✓   |  ✓   |           | 41.67 ± 1.03   | 31.51 ± 0.50   |
> |   ✓    |         |    ✓   |  ✓   |     ✓     | 53.41 ± 1.56   | 35.12 ± 0.80   |
> |        |    ✓    |    ✓   |  ✓   |           | 45.31 ± 2.68   | 34.80 ± 1.68   |
> |        |    ✓    |    ✓   |  ✓   |     ✓     | 54.99 ± 1.73   | 36.82 ± 0.88   |
>
>
> 2. Upper bound:
>    - (Response) Thank you for the suggestion. We agree and have updated our captions, tables, and other content to fix this.
>
> &nbsp;
>
> Strengthen:
>
> 3. Update $S_c$ to $S_{c+1}$
>    - (Response) The subscript c represents the class c of synthetic data and hence updating it to c+1 would not be correct. To make it clearer we have updated the description of the Algorithm.
>
> 4. Failure of MTT
>    - (Response) We observe that MTT runs out of memory for visual data distillation across various datasets and images per class (IPC) settings. This scaling problem of MTT has been extensively examined in prior research [1,2,3]. The objective function of MTT requires unrolling several (for eg. T) stochastic gradient descent (SGD) updates with synthetic images and aligning the resulting model weights with those obtained by training on the original dataset. This unrolling of T optimization steps necessitates backpropagation through T gradient computational graphs. Consequently, this approach requires substantial GPU memory, making it impractical for larger datasets. We have also briefly added this to Section 4.2 of our paper.
>
> 5. Discrepancy gap in Figure 3 and Figure 5
>    - (Response) We would like to clarify the difference between Figure 3 and Figure 5. The confusion may arise due to the use of different video (image-audio pairs) in Figure 3 for real and distilled data.
>    - The Distribution Matching method initializes distilled data with real data and then, during training, randomly selects real batches of image-audio pairs. Our matching losses are then used to backpropagate and update the distilled pairs. Figure 3 shows the training at an intermediate state, so the real batch used for matching in that iteration differs from the initialized distilled pairs. In contrast, Figure 5 visualizes how the initialized distilled data (epoch 0) appears after training (epoch 5000) across different images-per-class settings.
>
> &nbsp;
>
> Minor:
>
> 6. Table formatting
>     - (Response) Thank you for the suggestion. In the final revision, we will adjust the tables to ensure more consistent formatting, taking into account the varying sizes and spaces, along with incorporating all the reviewers' feedback.
>
> [1] Cui et al. “DC-BENCH: Dataset condensation benchmark”, NIPS 2022 \
> [2] Zhou et al. “Dataset distillation using neural feature regression”, NIPS 2022 \
> [3] Cui et al. “Scaling up dataset distillation to imagenet-1k with constant memory”, PMLR 2023

---

### Review · Reviewer_JgdC · 2024-09-15

**Summary Of Contributions:**

The paper extends the unimodal dataset distillation into audio-visual dataset distillation, aiming to obtain smaller synthetic datasets while preserving the cross-modal semantic association between audio and visual data. Based on distribution matching (DM) technique, the authors introduce two novel loss functions, implicit cross-matching (ICM) and cross-modal gap matching (CGM), to enhance the alignment between audio and visual modalities. The method is validated through sufficient experiments on four audio-visual datasets, showing significant improvements over unimodal distillation methods.

**Audience:**

No

**Broader Impact Concerns:**

Concerns are addressed in the Broader Impact Statement section.

**Claims And Evidence:**

Yes

**Requested Changes:**

Critical changes:

1. Please clarify why the performance of VGGSound is weak in the paper.

2. It would be better to discuss the scalability of the method to large-scale datasets in the paper.

Strengthens:

3. Suggested formatting improvements:

    * The sentence starting with "While dataset distillation techniques..." is long and complex. Consider splitting it into two sentences or reconstructing it for clarity.

    * The caption of Figure 15 overlaps with the page number "24". This should be adjusted.

**Strengths And Weaknesses:**

Strengths:
1. The paper introduces the concept of audio-visual dataset distillation, which is a new and relatively unexplored domain.
2. The authors propose various methods such as the cross-modal losses to adapt distribution matching to audio-visual task, which are novel and valuable.
3. The experiments span across four major datasets, providing convincing evidence of the method’s effectiveness. The ablation study is also a strength, as it carefully isolates and evaluates the impact of different components in the method (e.g., novel loss functions, herding-based initialization).

Weaknesses:

1. In Table 3, the performance on VGGSound is notably poor. Could this be due to the larger number of classes in VGGSound compared to other datasets? This raises concerns about the scalability of the proposed method on datasets with a high number of classes.

2. Audio-visual classification models typically achieve over 50% accuracy on VGGSound (e.g., [1] reports 65% accuracy). Why does the model trained on the whole dataset in this paper only achieve 25.54% accuracy? This discrepancy requires further clarification.

3. In Related Work, it is unclear why the method proposed by Wu et al. (2023) cannot be extended to audio-visual dataset distillation. Can the authors provide a more detailed explanation? Additionally, can the proposed method for audio-visual distillation be adapted for the image-text domain?

4. According to Fig. 5, when IPC is large, the synthetic images and audio appear quite similar to the original samples. This contrasts with MTT, where distilled images remain highly abstract and may contain information from multiple sources. This raises concerns about whether the proposed method will be effective on large-scale datasets with many concepts that need to be represented in a limited number of samples.

---

> ### Author Response · Authors · 2024-09-20
> **Response to the requested changes**
>
> We sincerely thank the reviewer for their valuable comments and insightful suggestions! We have carefully addressed your concerns in the revised paper as detailed below.
>
> 1. Performance on VGGSound (Weakness 1&2  and Requested changes 1&2)
>   - Distilled data performance:
>     - Due to the large number of categories in VGGSound, even training on the full original dataset yields lower performance compared to smaller datasets with fewer classes.
> Although our approach demonstrates better scalability and outperforms the baselines on VGGSound, the overall performance remains limited. This scenario is similar to ImageNet-1K for visual-only dataset distillation [1] performed poorly and required additional techniques like soft label alignment and reduced memory usage [2]  to enhance performance. These extensions are outside the scope of our current work, and we aim to tackle them in the future.
>     - However, we would also like to point out that the performance gap of VGGSound between Ours and the whole data is similar to the other datasets. For eg. (in Tab. 3) for IPC 20, the absolute difference between Ours and whole data is \~15\% on VGGSound, which is similar to MUSIC-21(\~15\%) and AVE(\~12\%).
>     - Hence, our approach is scalable with distilled data sizes and target dataset sizes and our future attempts will explore to bridge this performance gap (as previously highlighted in the discussion section).
>
> - Whole data performance:
>   - We follow established practices in dataset distillation research and quote the performance on the whole dataset as a reference.  The performance of whole data is lower than recent state-of-the-art audio-visual recognition works [3, 7]. This is primarily because our training architecture is intentionally smaller, utilizing fewer convolutional layers and a simple ensemble-based fusion. In contrast, [3] employs a deep transformer model with a fusion bottleneck, allowing for more intricate fusion strategies. In dataset distillation, the goal is to condense large datasets into much smaller ones. Complex networks, however, tend to overfit when trained on such small distilled datasets [8], making simpler architectures more suitable in this scenario.
>   - This difference in whole data performance is similar to previous works on visual-only dataset distillation. For example, on ImageNet-1k whole data performance is 33% [2] while the SOTA on ImageNet classification is ~92% [4]. As dataset distillation is a relatively new field, future research is needed to address this performance discrepancy.
> - We have updated our Discussion section and experiment section.
>
> 2. Extension of Vision-language dataset distillation (Wu et al. (2023))[5] (Weakness 3):
> - Thanks for the suggestion. A straightforward extension of [5] is not possible due to:
>    - Difference in modality : Wu et al. uses vision and text, however we use vision and audio and unlike text, audio data is temporally synchronized with visual components, expresses much variability, and is commonly represented as spectrograms of rich, isolated time-frequency patterns.
>    - Difference in Task: the main task of Wu et al. (2023) is instance-based image-to-text retrieval and text-to-image retrieval, however our task is class-based audio-visual classification and classwise audio-visual retrieval. Hence their approach is to learn distilled data that is instance-wise paired, while our distilled data is paired classwise.
>    - Scalability of their method: They extend MTT approach for their instance-wise distilled data pairs, which has proved to be difficult to scale to higher image-per-class setting in our experiments due to it’s higher memory requirement.
>    - We have updated our related works to highlight this.
>  - With some modifications, our approach can be adopted for image-text domain. Since image-text pairs have no class-labels, we can treat all pairs belonging to a single class and use our distillation losses. This would help us distill image and text data that are individually aligned and also aligned with each other with fairly covering the data distribution. Additionally, we would require pairs-wise matching losses to ensure that each image-text pair is correctly matched while being distinguished from other pairs.

---

> > ### Author Response · Authors · 2024-09-20
> > **Response to the requested changes (continued)**
> >
> > 3. Synthetic images for large IPC (Weakness 4):
> >    - We would like to clarify that the abstractness of the distilled data depends on several factors like image resolution and number of images per class and may not be related to the performance.
> >    - In our work, we use an image resolution of 224×224, whereas MTT employs a maximum resolution of 128×128 on the ImageNet subset. This is evident in Wu et al.[5], who also use a 224×224 resolution and achieve the best results, even though their distilled images closely resemble the original images. Also, an extension of MTT [6] which improves performance beyond whole data shows that using MTT would also result in with very less abstraction in images at high image-per-class settings (Fig 5 of [6]).
> >
> > 4. Formatting improvements(Requested changes 3):
> >    - Thank you for your suggestion. We have updated our manuscript to reflect this.
> >
> > Thank you again for your valuable feedback. We welcome any further questions or suggestions you may have.
> >
> > [1] Zhou et al., “Dataset distillation using neural feature regression”, NIPS 2022\
> > [2] Cui et al., “Scaling up dataset distillation to imagenet-1k with constant memory”, PMLR 2023\
> > [3] Nagrani et al., “Attention bottlenecks for multimodal fusion”, NIPS 2021\
> > [4] Srivastava et al., “Omnivec: Learning robust representations with cross modal sharing”, WACV 2024\
> > [5] Wu et al.,”Vision-Language Dataset Distillation”, TMLR 2024\
> > [6] Guo et al., “Towards lossless dataset distillation via difficulty-aligned trajectory matching”, ICLR 2024\
> > [7] Gong et al., “Contrastive audio-visual autoencoder”, ICLR 2023\
> > [8] Shiye Lei et al., “Dataset Distillation: A Comprehensive Review”, TPAMI 2023

---

### Author Response · Authors · 2024-08-10
**Paper revision**

We have revised the paper according to Reviewer s2fd’s comments. We would be happy to take any comments and suggestions from other reviewers to improve our paper further.

---

### Author Response · Authors · 2024-09-20
**Paper revision**

We have revised the paper according to Reviewer JgdC's comments. We would be happy to take any comments and suggestions from other reviewers to improve our paper further.

---

### Decision · Action_Editor_2APV · 2024-10-24

**Recommendation:** Accept as is

**Comment:**

The reviewers acknowledge that the authors have addressed most concerns, and specific aspects such as the use of Herding and Factor have been highlighted for their significant performance improvement. Additionally, the reviewers appreciate the insights into dataset distillation methods and notes that the revisions, including fixing typos and improving the content, have resolved prior issues.

**Audience:**

The topic of multimodal dataset distillation and the proposed method to bridge the semantic gap between visual and acoustic modalities is relevant to the broader field of machine learning, particularly for researchers and practitioners focused on multimodal learning and data distillation techniques.

**Claims And Evidence:**

Yes, the claims made in the submission are largely supported by accurate, convincing, and clear evidence.